# TRIP13 and APC15 drive mitotic exit by turnover of interphase- and unattached kinetochore-produced MCC

Dong Hyun Kim[1,2], Joo Seok Han[3], Peter Ly [1,2], Qiaozhen Ye[2], Moira A. McMahon[1,2,6], Kyungjae Myung[3,4], Kevin D. Corbett [2,5] & Don W. Cleveland[1,2]

The mitotic checkpoint ensures accurate chromosome segregation through assembly of the mitotic checkpoint complex (MCC), a soluble inhibitor of the anaphase-promoting complex/ cyclosome (APC/C) produced by unattached kinetochores. MCC is also assembled during interphase by Mad1/Mad2 bound at nuclear pores, thereby preventing premature mitotic exit prior to kinetochore maturation and checkpoint activation. Using degron tagging to rapidly deplete the AAA+ ATPase TRIP13, we show that its catalytic activity is required to maintain a pool of open-state Mad2 for MCC assembly, thereby supporting mitotic checkpoint activation, but is also required for timely mitotic exit through catalytic disassembly of MCC. Strikingly, combining TRIP13 depletion with elimination of APC15-dependent Cdc20 ubiquitination/degradation results in a complete inability to exit mitosis, even when MCC assembly at unattached kinetochores is prevented. Thus, mitotic exit requires MCC produced either in interphase or mitosis to be disassembled by TRIP13-catalyzed removal of Mad2 or APC15-driven ubiquitination/degradation of its Cdc20 subunit.

[1] Ludwig Institute for Cancer Research, San Diego Branch, La Jolla, CA 92093, USA. [2] Department of Cellular and Molecular Medicine, University of California-San Diego, La Jolla, CA 92093, USA. [3] Center for Genomic Integrity, Institute for Basic Science (IBS), Ulsan 44919, Republic of Korea. [4] School of Life Sciences, Ulsan National Institute for Science and Technology (UNIST), Ulsan 44919, Republic of Korea. [5] Department of Chemistry, University of California-San Diego, La Jolla, CA 92093, USA. [6]Present address: Ionis Pharmaceuticals, 2855 Gazelle Ct, Carlsbad, CA 92010, USA. Correspondence and requests for materials should be addressed to K.D.C. (email: kcorbett@ucsd.edu) or to D.W.C. (email: dcleveland@ucsd.edu)

The mitotic checkpoint (or spindle-assembly checkpoint) delays the onset of anaphase until all chromosomes have successfully attached to spindle microtubules, thereby serving as the primary guard against chromosome missegregation. Anaphase onset is controlled by a multisubunit E3 ubiquitin ligase, the anaphase-promoting complex/cyclosome (APC/C)[1–4]. The APC/C, together with its co-activator Cdc20, targets cyclin B and securin for proteasome-mediated destruction, thereby causing exit from mitosis and activation of separase [reviewed in ref. [5]]. The mitotic checkpoint functions through unattached kinetochores, which generate a soluble mitotic checkpoint complex (MCC) consisting of Mad2, Cdc20, BubR1, and Bub3. The MCC binds the APC/C$^{Cdc20}$ complex as a substrate analog and directly inhibits its activity[6–8]. While MCC is mainly produced in mitosis by the mitotic checkpoint, low levels of MCC are also produced in interphase cells, and this pool of MCC is thought to be important for inhibiting early mitotic exit prior to kinetochore maturation and mitotic checkpoint activation[9–12].

The key component controlling MCC assembly is Mad2, an essential checkpoint protein containing a HORMA domain[13] that can adopt two different conformations: "open" (O-Mad2) or "closed" (C-Mad2). C-Mad2 binds partner proteins, including Cdc20 and Mad1, through a distinctive "seat belt" interaction, in which the Mad2 C-terminus embraces a short Mad2-interacting motif (MIM) in the partner. In O-Mad2, the C-terminal seat belt occupies the MIM binding site to prevent complex formation[14–17]. In vivo and in vitro evidence has shown that O-Mad2 is recruited to Mad1:C-Mad2 complexes at unattached kinetochores, where it is converted to C-Mad2 and assembled with Cdc20 to form the core of the mitotic checkpoint complex[18–22]. After assembly of the Cdc20:C-Mad2 complex, recruitment of BubR1 and Bub3 complete assembly of MCC, which binds to APC/C$^{Cdc20}$ and inhibits its recognition of cyclin B and securin, thereby delaying mitotic exit. Low-level MCC assembly in interphase is thought to occur through a similar mechanism, catalyzed by Mad1:C-Mad2 localized to nuclear pores[11,23].

Once all kinetochores have become attached to microtubules, the mitotic checkpoint is silenced and APC/C$^{Cdc20}$ is reactivated for cyclin B and securin ubiquitination and their subsequent proteasomal destruction, which in turn initiates anaphase onset. At each kinetochore, stable microtubule attachment releases the Mad1:Mad2 complex[24–27] and PP1 phosphatase activity is elevated to inactivate checkpoint kinases, including Mps1, thereby leading to suppression of additional MCC production[28–33]. Existing APC/C$^{Cdc20}$-bound MCC (APC/C$^{Cdc20-MCC}$) must then be disassembled and/or degraded to reactivate APC/C$^{Cdc20}$ recognition of cyclin B and securin. One proposed mechanism for APC/C$^{Cdc20}$ reactivation is ubiquitination of the Cdc20 subunit of MCC by the inhibited APC/C$^{Cdc20}$ itself, which is enabled by a conformational change within the APC/C allowing recruitment of an E2 enzyme and ubiquitination of Cdc20 in MCC[34–37]. Loss of APC15, a 14 kD APC/C subunit which is dispensable for APC/C assembly and activity[38–40], eliminates this reactivation pathway by preventing the conformational change required for E2 recruitment and Cdc20 ubiquitination[41,42]. A second pathway potentially driving mitotic exit has been proposed to involve the AAA+ ATPase TRIP13 acting with an adapter protein p31$^{comet}$ to extract Mad2 from the assembled MCC. p31$^{comet}$ specifically binds C-Mad2 alone or in complex with Cdc20 and other MCC subunits[43–46]. TRIP13 and p31$^{comet}$ together can mediate disassembly of MCC complexes in vitro[47–49], acting by partially unfolding the N-terminus of C-Mad2 to catalyze its conversion to O-Mad2[50–52]. Whether and how the two proposed pathways for MCC disassembly/degradation— APC15-mediated ubiquitination/degradation and TRIP13-mediated disassembly—synergize to mediate APC/C$^{Cdc20}$ activation and mitotic checkpoint silencing is not known.

Despite its proposed role in MCC disassembly and checkpoint silencing, deletion or complete loss of TRIP13 in C. elegans[53] or human cells[54–56] causes defects in checkpoint activation, rather than failure of checkpoint silencing. In the absence of TRIP13, most soluble Mad2 is in the closed conformation, rendering it unable to be recruited to unattached kinetochores and assembled with Cdc20 into the MCC[54]. This finding, plus the observation that purified O-Mad2 spontaneously converts to C-Mad2 in vitro over the course of several hours[17], suggests that TRIP13 is required for mitotic checkpoint activation by maintaining a pool of O-Mad2.

The proposed dual roles of TRIP13 in the mitotic checkpoint mirror a similar duality in its observed roles in cancer. Recently, it was shown that biallelic loss-of-function mutations in TRIP13 underlie mosaic variegated aneuploidy and Wilms tumor in children, establishing TRIP13 as a bona fide tumor suppressor[55]. Conversely, several studies have highlighted that TRIP13 is overexpressed in many cancer types, and that high-level TRIP13 expression correlates with poor patient outcomes[57,58]. The contributions of TRIP13 to tumorigenesis and cancer progression remain largely mysterious.

Here, we develop an inducible system to rapidly and completely deplete TRIP13 in cultured human cells by engineering auxin-inducible degron (AID) tags into both endogenous TRIP13 loci. Loss of TRIP13 during interphase is shown to trigger checkpoint activation failure in the next mitosis due to a loss of soluble O-Mad2. Conversely, cells that enter mitosis soon after depletion of TRIP13, prior to full O-Mad2 to C-Mad2 conversion, show a dramatic checkpoint-mediated delay in mitotic exit, revealing TRIP13's critical role in checkpoint silencing. Further, we show that checkpoint silencing involves TRIP13 acting synergistically with APC15-mediated Cdc20 ubiquitination to disassemble or degrade, respectively, soluble and APC/C$^{Cdc20}$-bound MCC. Strikingly, we find that mitotic exit requires MCC produced either in interphase or mitosis to be disassembled by TRIP13-catalyzed removal of Mad2 or APC15-driven ubiquitination/degradation of its Cdc20 subunit.

## Results

**Long-term TRIP13 depletion compromises cell fitness.** To determine the roles of TRIP13 in the mitotic checkpoint, we sought to develop a means of acutely controlling TRIP13 levels in live cells. We used CRISPR/Cas9 to sequentially introduce sequences encoding green fluorescent protein (GFP) and an auxin-inducible degron (AID)[59,60] into both TRIP13 alleles of nearly diploid human DLD-1 cells (hereafter, TRIP13$^{AID}$) also stably expressing a fluorescent histone H2B protein (H2B-mRFP) and the TIR1-9Myc E3 ubiquitin ligase to permit inducible ubiquitination of AID-tagged proteins (Fig. 1a). Upon addition of indole-3-acetic acid (IAA), GFP-AID-TRIP13 was rapidly degraded with a $t_{1/2}$ of ~7 min (Fig. 1b, Supplementary Fig. 1A), becoming undetectable (to <3% of its initial level) within 30 min. No discernable difference in growth between normal or TRIP13-depleted cells was seen after 3 days (Supplementary Fig. 1B). We first used time-lapse live-cell imaging (using H2B-mRFP to monitor chromosome condensation and decondensation) to examine the fate of cells in the first mitosis after TRIP13 depletion. Addition of nocodazole to inhibit spindle microtubule assembly caused a chronic mitotic arrest in TRIP13-containing cells (Fig. 1c, left panel), whereas most cells entering mitosis ≥24 h after TRIP13 depletion exited mitosis within 40 min (Fig. 1c; right panel). In unperturbed mitoses (no nocodazole), the time from nuclear envelope breakdown (NEBD) to chromosome alignment in metaphase

was indistinguishable in the presence or absence of TRIP13 (~23 min, Fig. 1d; left panel), but in the absence of TRIP13 the transition from metaphase-to-anaphase was accelerated (lowering it from ~20 to ~11 min; Fig. 1d, right panel). Normal metaphase-to-anaphase timing could be rescued in TRIP13-depleted cells by expression of wild-type TRIP13, but not with TRIP13 bearing the Walker B motif E253Q mutation (TRIP13[EQ]), which allows for ATP binding and stabilizes the TRIP13 hexamer, but eliminates ATP hydrolysis[50,52] (Supplementary Fig. 1C-F). These findings demonstrate that TRIP13 is not strictly needed for entry into mitosis or for chromosome congression, but is required for activation and/or maintenance of the mitotic checkpoint.

Live-cell imaging of TRIP13[AID] cells revealed significant errors in chromosome segregation during anaphase of the first mitosis

after TRIP13 depletion (Supplementary Fig. 1G) that drove chromosomal aneuploidy in the daughter cells (as determined by fluorescence in situ hybridization (FISH) analyses—Fig. 1e). Clonogenic assays indicated that prolonged (3 week) depletion of TRIP13 resulted in significantly smaller colony size compared to TRIP13[WT] cells (Supplementary Fig. 1H). Collectively, this evidence demonstrates that TRIP13 normally acts to prevent aneuploidy and that loss of TRIP13 compromises long-term cell fitness.

**TRIP13 maintains open Mad2 to support checkpoint activation.** To examine Mad2 conformation in the absence of TRIP13, we depleted TRIP13 in TRIP13[AID] cells by addition of IAA, and examined Mad2 conformation in cell lysates by Mono-Q ion-exchange chromatography (which separates the two Mad2

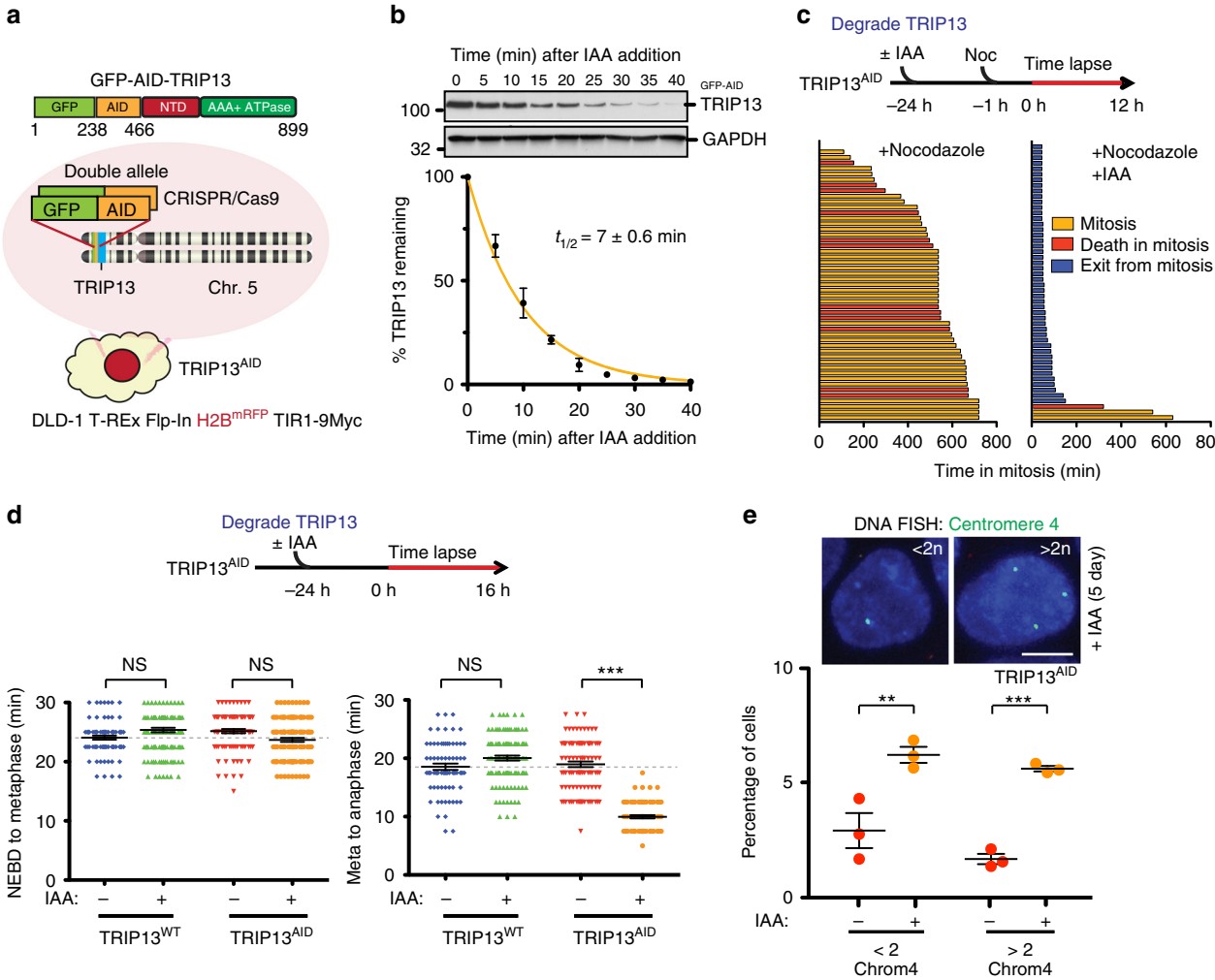

**Fig. 1** Abrupt TRIP13 depletion compromises mitotic checkpoint activation. **a** Schematic illustration of CRISPR-Cas9 mediated genome editing of DLD-1 T-REx Flp-In H2B[mRFP] + TIR1-9Myc[+] human colon cancer cell line to tag both endogenous TRIP13 alleles with green fluorescent protein (GFP) and an auxin-inducible degron (AID). **b** Upper: Indole-3-acetic acid (IAA) was added to engineered TRIP13[AID] cells and level of GFP-AID-TRIP13 was measured by quantitative immunoblot analysis. Lower: Degradation kinetics of GFP-AID-TRIP13 from triplicate measurements, fit using a single-exponential decay function (yellow line). **c** TRIP13 function in mitotic checkpoint. Upper: Schematic of experiment to test the role of TRIP13 in mitotic checkpoint activation. Lower: Timing of mitotic exit and cell fate as measured by time-lapse live-cell imaging. ($n = 57$ for +Nocodazole, $n = 54$ for +Nocodazole, +IAA). **d** TRIP13 function in unperturbed mitosis. Upper: Schematic of experiment to test the contribution of TRIP13 in mitotic checkpoint activation. Lower: For each cell, timing of nuclear envelope breakdown (NEBD) to metaphase (left panel) and metaphase-to-anaphase (right panel) were measured. Cells monitored: $n = 78, 137, 100, 120, 79, 129, 102, 81$. **e** Upper: Representative images of TRIP13[AID] cells treated with IAA for 5 days and probed by FISH for the centromere of chromosome 4. Lower: Percentage of cells with the indicated number of chromosome 4 centromere foci after 5-day IAA treatment. Total cells counted: 1156 for control, 1119 for IAA treatment over three separate experiments. Scale bar, 5 μm. $P$-values for **d** and **e** were calculated using an unpaired two-tailed t-test (**$P < 0.01$, ***$P < 0.001$, NS not significant). Lines in **b**, **d**, and **e** represent the mean ± s.e.m.

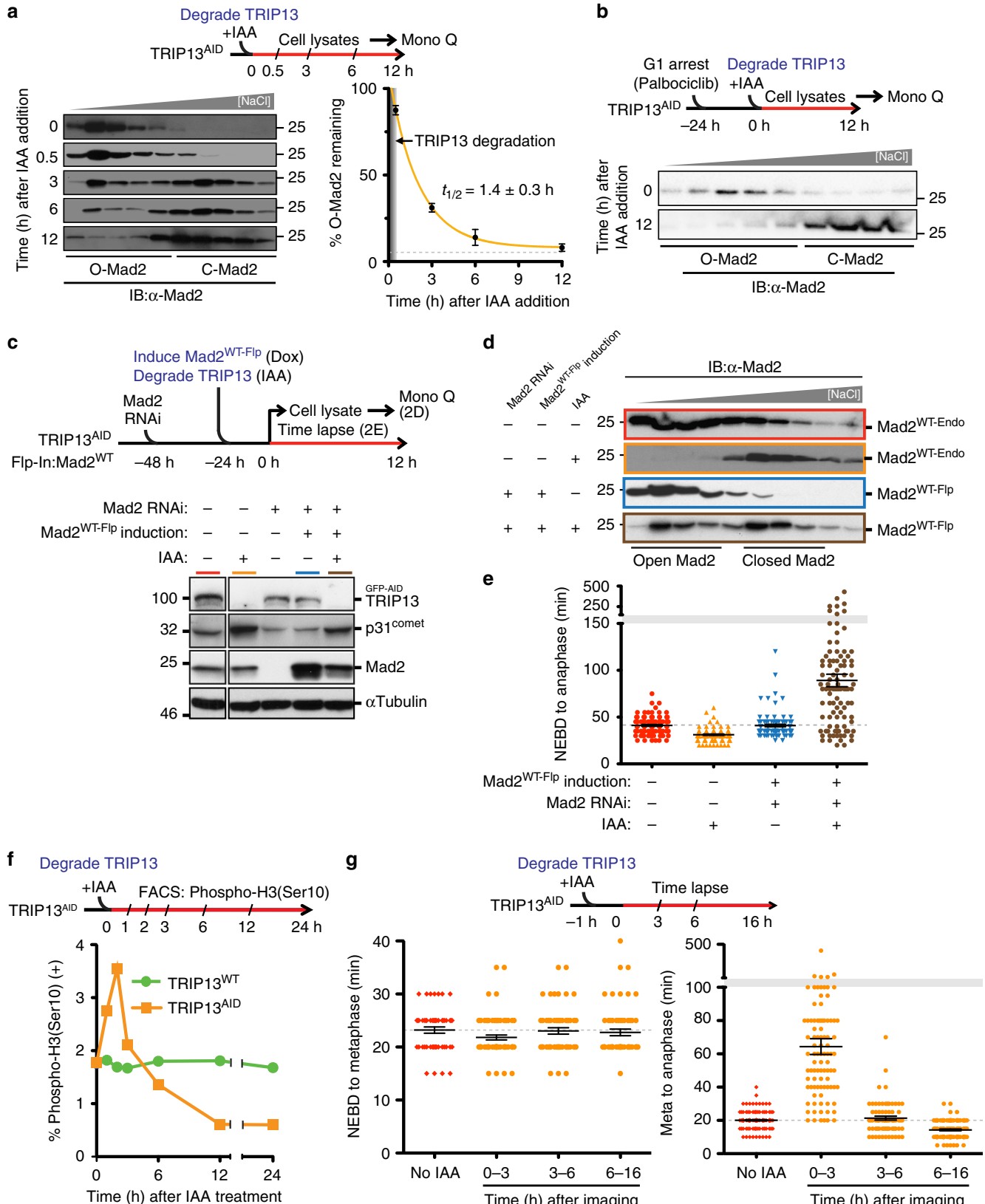

conformers[16,17,50,52]). Mad2 was mostly in its open conformer prior to TRIP13 depletion, in agreement with prior studies[17] (Fig. 2a). Over time, however, Mad2 gradually converted to the closed conformation, with a conversion half-time of ~1.4 h and reaching an equilibrium at ~8% O-Mad2 (Fig. 2a, right panel).

This rate of conformational conversion is consistent with the in vitro rate of spontaneous O-Mad2 to C-Mad2 conversion (half-time; 1.5–2 h)[61] and with the equilibrium percentage of O-Mad2 seen in our earlier in vitro measurements[50]. These data clearly indicate that a major role of TRIP13 is to counteract

**Fig. 2** TRIP13 regulates mitotic checkpoint activation and silencing through Mad2. **a** Upper: Schematic of experiment to test kinetics of Mad2 conformational conversion after TRIP13 depletion. Lower left: Mono-Q ion-exchange chromatography analysis of soluble Mad2 in TRIP13[AID] cells after IAA treatment for the indicated times. Lower right: Kinetics of O-Mad2 to C-Mad2 conversion was calculated from triplicate measurements, fit using a single-exponential decay function (yellow line). $t_{1/2} = 1.4 \pm 0.3$ h, plateau = 8% O-Mad2 (gray dashed line). **b** Mad2 conformation in G1 after TRIP13 depletion. TRIP13[AID] cells were arrested in G1 with Palbociclib and Mad2 conformation was measured by Mono-Q ion-exchange chromatography at 0 or 12 h after addition of IAA to degrade TRIP13. **c** Upper: Schematic of the approach to replace endogenous Mad2 in TRIP13-depleted cells (used in **d** and **e**). Lower: Immunoblot analysis of the replacement of endogenous Mad2 with Mad2[WT-Flp] induced by doxycycline. **d** Mono-Q ion-exchange chromatography analysis of Mad2 conformation after replacement of endogenous Mad2 with Mad2[WT-Flp], in the presence (blue box) or absence (brown box) of TRIP13. High-level expression of Mad2[WT-Flp] resulted in a significant O-Mad2 population even after TRIP13 depletion. **e** Mitotic timing (NEBD-to-anaphase) after replacement of endogenous Mad2 with Mad2[WT-Flp], measured by time-lapse live-cell imaging. ($n = 100$ for each condition). **f** Upper: Schematic of experiment to measure population of mitotic TRIP13[WT] and TRIP13[AID] cells upon IAA treatment. Lower: Percentage of phospho-histone H3 (Ser10) positive cells was measured using flow cytometry at different time points after IAA treatment. Detailed FACS profiles in Supplementary Fig. 2D. **g** Upper: Schematic of experiment to measure mitotic timing in TRIP13[AID] cells upon IAA treatment. Lower: NEBD-to-metaphase (left) and metaphase-to-anaphase (right) timing of TRIP13[AID] cells at different time points after IAA addition. (From left to right panel, $n = 50, 69, 56, 58, 176, 93, 70,$ and 110). Lines in **e** and **g** represent the mean ± s.e.m.

spontaneous conversion of O-Mad2 to C-Mad2, the latter of which cannot be recruited to unattached kinetochores and assembled into the MCC. Importantly, this conversion was not related to cell-cycle progression: in cells arrested in G1 (by addition of the Cdk4/6 inhibitor palbociclib[62]), >90% of Mad2 converted to the closed conformation within 12 h of TRIP13 depletion (Fig. 2b).

Cells chronically depleted of TRIP13 have been reported to show changes in overall Mad2 and p31[comet] protein levels compared to normal cells[54,55]. We observed similar changes upon TRIP13 depletion, including a ~2-fold reduction in Mad2 level and a >5-fold increase in p31[comet] levels (Supplementary Fig. 3). When we replaced endogenous Mad2 with a mutant (Mad2[H191A]) that predominantly adopts the open conformation even in the absence of TRIP13[52,63], we did not observe increased p31[comet] levels upon TRIP13 depletion (Supplementary Fig. 3F). Thus, p31[comet] is stabilized by binding C-Mad2 and strongly accumulates in the absence of TRIP13[54].

To test whether the checkpoint activation defect upon longer-term TRIP13 depletion was caused by insufficient O-Mad2, we stably integrated a gene encoding untagged, wild-type Mad2 into a doxycycline-inducible FRT locus in the TRIP13[AID] cell line (hereafter Mad2[WT-Flp]). We then depleted endogenous Mad2 by siRNA and induced the Mad2[WT-Flp] gene (whose mRNA is resistant to the siRNA) (Fig. 2c). In the presence of TRIP13, most Mad2 produced from either the endogenous or Mad2[WT-Flp] genes was in the form of O-Mad2 (Fig. 2d) and the cells traversed mitosis with normal timing (Fig. 2e). In contrast to the nearly complete conversion of endogenous Mad2 (Mad2[WT-endo]) to the closed form after TRIP13 degradation (Fig. 2d), doxycycline-mediated synthesis of new Mad2 resulted in a significant pool of soluble O-Mad2 even in the absence of TRIP13 (Fig. 2d), which in turn supported a strong delay in the timing of mitotic exit in these cells (Fig. 2e). Thus, newly synthesized Mad2 adopts the O-Mad2 conformation and this new Mad2 is sufficient to rescue mitotic checkpoint activation in the absence of TRIP13, while the subsequent delay in mitotic exit supports a role for TRIP13 in MCC disassembly to permit anaphase onset.

**TRIP13 has a critical role in mitotic checkpoint silencing.** In vitro, TRIP13 and its Mad2-binding adapter protein p31[comet] have been shown to convert C-Mad2 to O-Mad2[50] and to release Mad2 from Cdc20:C-Mad2 or the four-protein MCC[47,48,64]. These findings, plus reports that RNAi-mediated reduction of TRIP13 can slow mitotic exit in some cells[49], have led to the idea that TRIP13 and p31[comet] participate in checkpoint silencing. We initially tested a role for TRIP13 in checkpoint silencing by

measuring the mitotic index (by scoring histone H3 serine 10 phosphorylation-positive cells) over time after inducing TRIP13 degradation. This revealed a sharp, transient increase in mitotic cells between 1 and 3 h after TRIP13 degradation, consistent with a TRIP13 role in mitotic checkpoint silencing and timely mitotic exit (Fig. 2f). We then used time-lapse live-cell imaging to determine that, despite unaffected NEBD-to-metaphase timing, metaphase-to-anaphase timing was significantly delayed (more than tripling the typical 20 to 64 min) in cells entering mitosis between 0 and 3 h after TRIP13 degradation (Fig. 2g). Cells entering mitosis 3–6 h after TRIP13 degradation exited mitosis with nearly wild-type timing, while cells entering mitosis even later (6–16 h after TRIP13 degradation) exited mitosis faster than wild-type cells (Fig. 2g), consistent with failure of mitotic checkpoint activation and with the mitotic timing we observed after 24-h TRIP13 depletion (Fig. 1d) and in TRIP13[−/−] cells[54].

To more clearly establish that the mitotic exit delay we observed shortly after TRIP13 depletion was due to a defect in mitotic checkpoint silencing, we next tested whether this delay depended on Mad2 and actively signaling kinetochores. We used siRNA to deplete Mad2 in TRIP13[AID] cells and then examined mitotic timing shortly after TRIP13 depletion. Mad2-depleted cells exited mitosis significantly faster than untreated cells regardless of whether or not TRIP13 was present (Supplementary Fig. 2A). We next prevented MCC assembly at unattached kinetochores by addition of reversine to inactivate the Mps1 kinase[65], and found that this treatment also accelerated mitotic exit both in the presence and absence of TRIP13 (Supplementary Fig. 2B). Together, these data indicate that the delay in mitotic exit that we observe shortly after TRIP13 depletion depends on both Mad2 and actively signaling kinetochores, supporting a direct role for TRIP13 in checkpoint silencing, most likely through MCC disassembly after kinetochore-microtubule attachment.

**TRIP13 and APC15 act in parallel to inactivate MCC.** Despite the strong delay in mitotic exit in cells that entered mitosis within 3 h of TRIP13 degradation, these cells did eventually exit mitosis. This observation indicated that MCC bound to APC/C[Cdc20] is either able to spontaneously disassemble, or that it is disassembled/degraded through an alternate pathway(s). Beyond TRIP13-mediated disassembly of Mad2 from MCC[48,49], another proposed pathway involves ubiquitination of Cdc20 within MCC bound to APC/C[Cdc20], resulting in Cdc20 degradation, MCC disassembly, and reactivation of APC/C[Cdc20] for recognition of cyclin B and securin[34-37]. For this second pathway, APC15, a 14

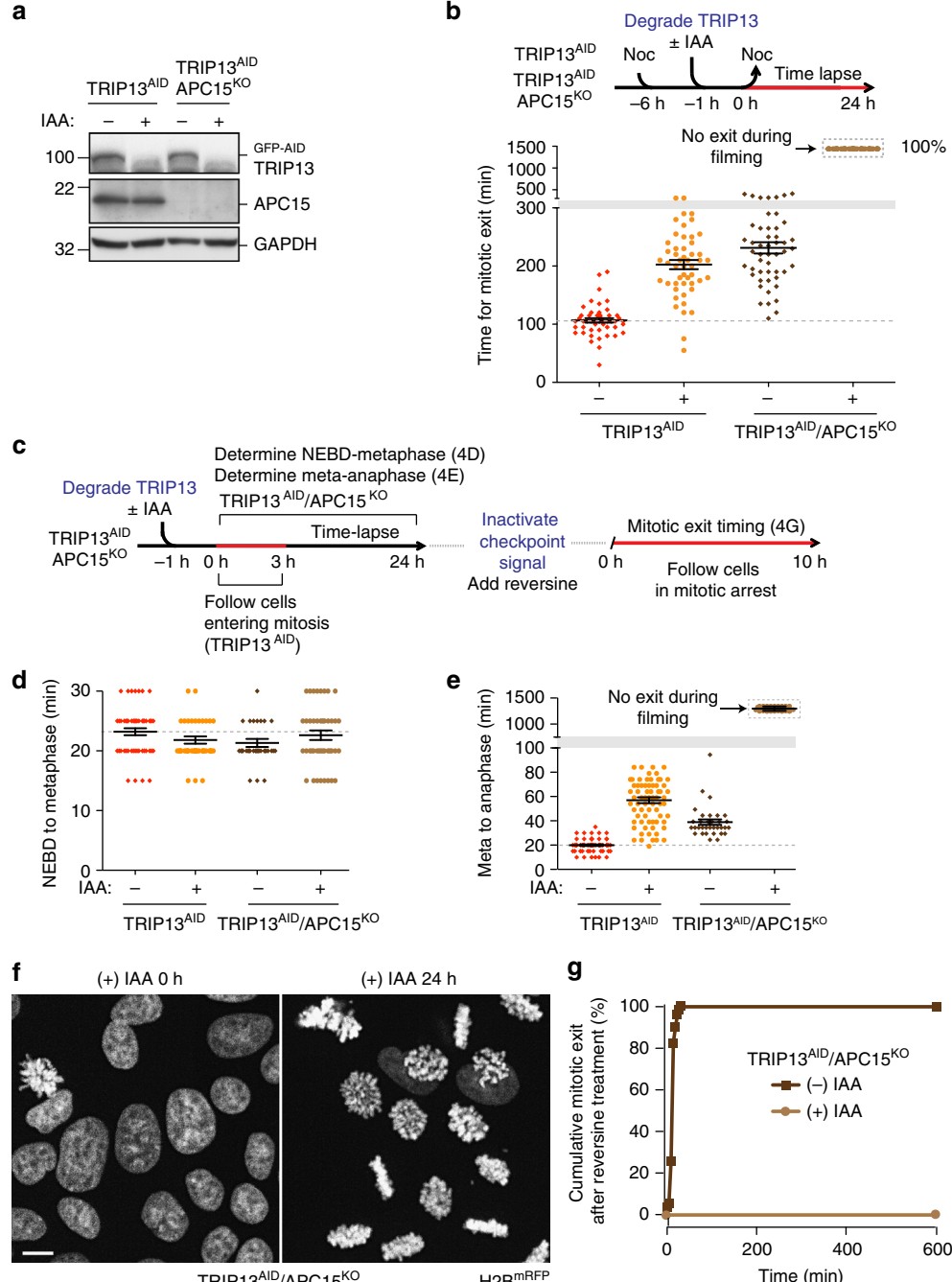

**Fig. 3** TRIP13 and APC15 synergistically act on kinetochore-generated MCC. **a** Western blot showing effective depletion of GFP-AID-TRIP13 in TRIP13[AID] and TRIP13[AID]/APC15[KO] cells. **b** Upper: Schematic of experiment to measure timing of mitotic exit after TRIP13 depletion by IAA addition. Lower: Timing of mitotic exit after 6-h nocodazole treatment, followed by release into DMEM. ($n = 50$ for each experimental condition). **c** Schematic of approach to monitor unperturbed mitosis after depletion of TRIP13 (by IAA addition) in TRIP13[AID]/APC15[KO] cells (used in **d**–**f**). **d** Unperturbed mitotic timing NEBD-to-metaphase; ($n = 50$ for each experimental condition). **e** Metaphase-to-anaphase of TRIP13[AID] and TRIP13[AID]/APC15[KO] cells entering mitosis 3 h after depletion of TRIP13. ($n = 70$ for each experimental condition). **f** Representative images of chromosomes in TRIP13[AID]/APC15[KO] cells after before IAA treatment (left) and 24 h after IAA treatment (right). Scale bar, 5 μm. **g** Mitotic exit timing after inhibition of kinetochore signaling. Reversine was added 24 h after the start of imaging to inactivate kinetochore signaling, and mitotic exit of cells previously arrested in mitosis was measured for 10 h. ($n = 70$ for each experimental condition). Lines in **b**, **d**, and **e** represent the mean ± s.e.m.

kDa APC/C subunit that is dispensable for overall APC/C[Cdc20] assembly and ubiquitination activity, has been implicated as a required factor for ubiquitination of Cdc20 in MCC-bound APC/C[Cdc20] through its promotion of a conformational change in APC/C, which allows recruitment of an E2 enzyme and ubiquitination of Cdc20 in MCC[41,42].

We tested whether mitotic exit in cells shortly after TRIP13 degradation depended on APC15-mediated Cdc20 ubiquitination and degradation. To do this, we inactivated both *APC15* alleles using CRISPR-Cas9 in TRIP13[AID] cells (Fig. 3a, hereafter, TRIP13[AID]/APC15[KO]). We treated cells with nocodazole to induce a mitotic checkpoint-mediated arrest, then released from

nocodazole after rapid depletion of TRIP13 with auxin-induced degradation, and tracked cell-cycle progression by live-cell microscopy. While cells depleted of either APC15 or TRIP13 showed delayed mitotic exit relative to wild-type cells, these cells did eventually exit mitosis (Fig. 3b). In contrast, 100% of cells depleted of both APC15 and TRIP13 were chronically arrested in mitosis with none escaping over 20 h of filming (Fig. 3b). We next tracked unperturbed mitosis in TRIP13[AID]/APC15[KO] cells, and found that while these cells had normal NEBD-to-metaphase timing and only a slight delay in mitotic exit when TRIP13 was present (Fig. 3d, e), degradation of TRIP13 by IAA treatment produced a durable metaphase arrest, with all cells remaining arrested for the duration of filming (21–24 h) (Fig. 3e, f). Finally, we treated mitotically arrested TRIP13[AID]/APC15[KO] cells with reversine to eliminate kinetochore production of MCC. We found that while cells retaining TRIP13 exited mitosis quickly, 100% of cells lacking both APC15 and TRIP13 (after IAA addition) remained mitotically arrested for at least 10 h (Fig. 3g). Thus, in the absence of the known catalytic pathways for MCC disassembly or degradation, cells are completely unable to reactivate APC/C[Cdc20-MCC] and exit mitosis, even in the absence of any kinetochore signaling.

**Mitotic exit relies on turnover of interphase-generated MCC.** Recently, multiple studies have indicated that in addition to being assembled at unattached kinetochores in mitosis, MCC may be generated at low levels outside of mitosis, likely by Mad1:Mad2 complexes localized at nuclear pores[10,11]. We tested if this pool of MCC produced in interphase might be enough to delay mitotic exit in the absence of the two catalytic MCC disassembly/degradation pathways. We depleted TRIP13 and simultaneously added reversine to eliminate kinetochore-dependent mitotic checkpoint signaling in TRIP13[AID]/APC15[KO] cells, and examined mitotic timing. Strikingly, 75% of cells lacking both APC15 and TRIP13 and which never activated the mitotic checkpoint remained arrested in mitosis for the duration of 24 h of filming (Fig. 4a). This prolonged mitotic arrest was dependent on both Mad2 and BubR1 (Fig. 4b, Supplementary Fig. 4C), as would be expected from a bona fide MCC-mediated delay. These data not only support the idea that a pool of MCC is synthesized in interphase independently of kinetochores, but also demonstrate that this interphase-produced pool of MCC is sufficient to cause sustained mitotic arrest in the absence of TRIP13 (and its catalytic MCC disassembly activity) or APC15 (and its stimulation of ubiquitination/degradation of the Cdc20 subunit of MCC).

In interphase, Mad1:Mad2 are bound to the nucleoplasmic face of the nuclear envelope through the TPR protein, and this pool of Mad1:Mad2 has been implicated in production of MCC during interphase[9,11,23]. To examine whether TPR (and its tethering of Mad1:Mad2 to the nuclear pore[11]) is essential for MCC generation by Mad1:Mad2 at the nuclear pore that is sufficient to trigger sustained mitotic arrest in the absence of TRIP13 and APC15, we used RNAi to deplete TPR from TRIP13[AID]/APC15[KO] cells (Supplementary Fig. 4,B) and monitored mitotic timing (Fig. 4c). Like TPR-containing cells (Fig. 4a), TRIP13[AID]/APC15[KO] cells entering mitosis after depletion of TPR (but competent for kinetochore-dependent mitotic checkpoint activation) were almost completely (98%) arrested after IAA-induced degradation of TRIP13 (Fig. 4c). However, when mitotic checkpoint signaling was prevented (by addition of reversine), reduced TPR levels sharply reduced the proportion of mitotically arrested cells [from 75% (Fig. 4a) to 28% (Fig. 4c)], consistent with TPR-dependent production of an APC/C[Cdc20] inhibitor.

Finally, we tested for the presence of MCC (either free or bound to APC/C[Cdc20]) in interphase TRIP13[AID]/APC15[KO] cells.

Cells were synchronized in either mid-G1 (by inhibiting CDK4/6 with Palbociclib) or in late G2 (by inhibiting CDK1 with RO-3306) and the levels of BubR1-bound Cdc20 and Mad2 determined by immunoprecipitation and immunoblotting (Supplementary Fig. 4D). In both G1 and late G2, low levels of APC/C-associated BubR1 and Mad2 were detected in extracts from cells containing TRIP13; much higher levels were found in extracts of cells depleted of TRIP13 (by IAA-induced degradation) (Fig. 4d). Overall MCC levels (as measured by BubR1-bound Mad2) also increased upon degradation of TRIP13 in G2-arrested cells (Supplementary Fig. 4D). These data support a role for TRIP13 in continual MCC disassembly throughout interphase as well as mitosis.

To test whether MCC is bound to APC/C during interphase when both TRIP13 and APC15 are missing, we monitored the level of MCC components associated with the core APC/C subunit APC3 in G1 or late G2-arrested TRIP13[AID]/APC15[KO] cells. While a small proportion of BubR1 was recruited to APC/C in the presence of TRIP13 in G1 cells (Fig. 4d), cells lacking APC15 and TRIP13 accumulated up to ~8-fold higher levels of APC/C-bound BubR1, Mad2, and Cdc20 (Fig. 4d). When combined with our evidence for chronic mitotic arrest in the absence of TRIP13, APC15, and mitotic checkpoint activation, these data provide strong evidence that MCC produced in interphase can quantitatively inhibit APC/C[Cdc20] and produce chronic mitotic arrest when TRIP13 and APC15 are absent.

## Discussion

Our use of rapid, inducible degradation of TRIP13 has demonstrated that Mad2 conformation in vivo is highly dependent on TRIP13 and its ATPase activity, with cellular O-Mad2 converted to C-Mad2 with a half-time of 1.4 h in the absence of TRIP13 (Fig. 2a). Since this matches the kinetics of spontaneous conversion of purified O-Mad2 to C-Mad2 in vitro[61], our evidence offers strong support that cellular O-Mad2 undergoes spontaneous conformational conversion in vivo that is independent of other factors. We also show that after loss of TRIP13, insufficient soluble O-Mad2 at the time of mitotic entry results in failure of mitotic checkpoint activation and that checkpoint activation can be rescued by re-introduction of additional O-Mad2. Together, these observations establish that a key function of TRIP13 throughout the cell cycle is to counteract the spontaneous conversion of O-Mad2 to C-Mad2, thereby maintaining a pool of soluble O-Mad2 which can be recruited to kinetochores in mitosis to produce MCC (Fig. 5) and clarify the dual roles of TRIP13 in mitotic checkpoint activation and silencing.

Our data also provide strong support for the idea that MCC is assembled outside of mitosis by Mad1, Mad2, and TPR, the latter of which tethers Mad1–Mad2 to the nuclear pore complex[11,23]. Our evidence suggests that APC15 and TRIP13 mediate continuous turnover of interphase-produced MCC complexes, which act both to restrain the activity of any APC/C[Cdc20] present in interphase, and provide a baseline level of pre-made MCC to delay mitotic exit prior to kinetochore maturation and mitotic checkpoint activation. Thus, TRIP13 and APC15 activity serve as a counter-balance to MCC assembly both in interphase and in mitosis[11]. The delicate balance between MCC assembly and disassembly/degradation pathways, and these pathways' coordination through space and time, will be important to consider as the contributions to cancer of both TRIP13 loss and overexpression are further studied.

Mitotic checkpoint silencing is initiated when spindle microtubules capture all kinetochores, thereby suppressing production of new MCC through delocalization of kinetochore-bound Mad1:C-Mad2[24–27] and reduced activity of kinetochore-bound

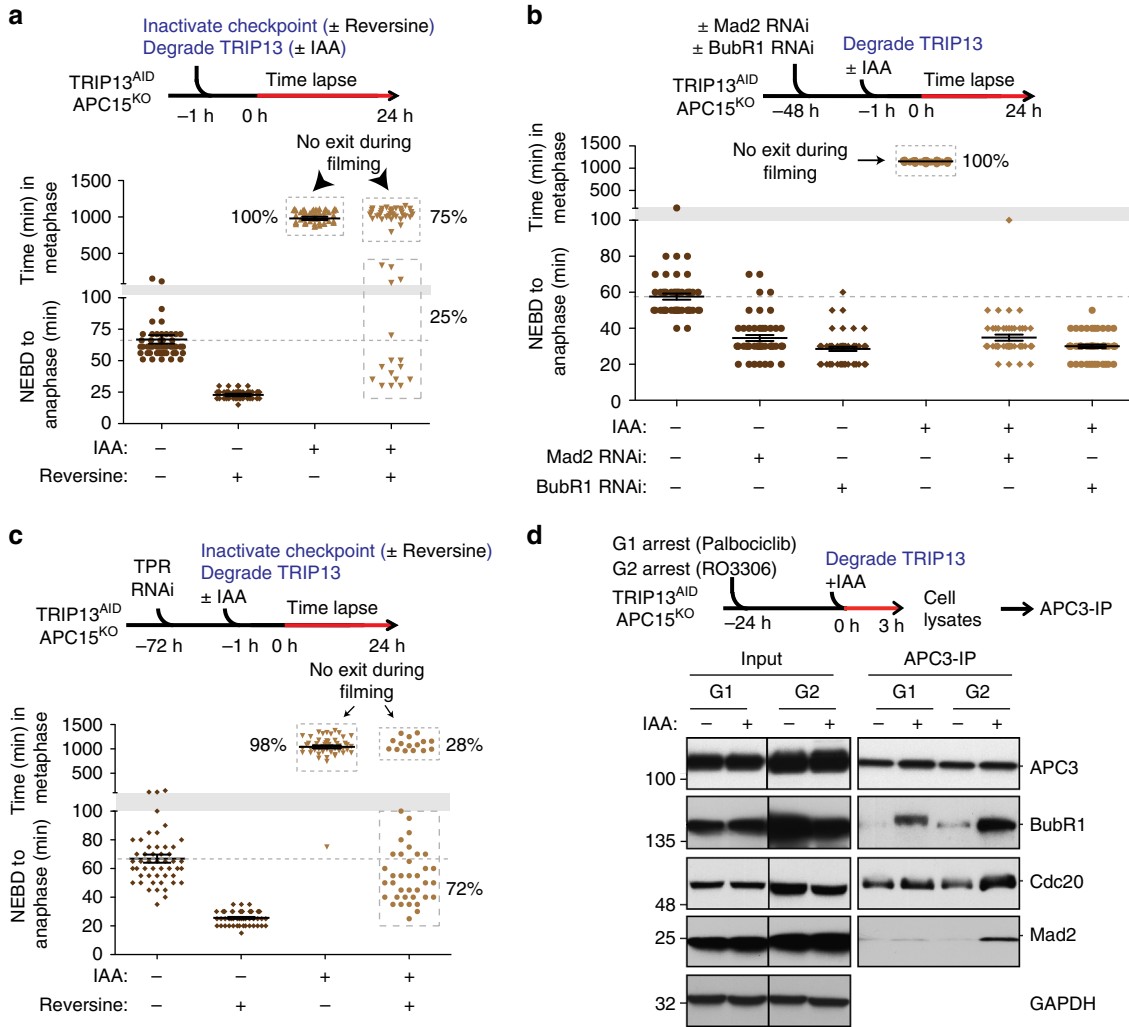

**Fig. 4** TRIP13 and APC15 are required for turnover of MCC produced in interphase. **a** Upper: Schematic of experiment to measure the dependence of the observed mitotic exit delay on kinetochore-catalyzed MCC production. Lower: mitotic timing in reversine-treated TRIP13[AID]/APC15[KO] cells in the presence or absence of IAA. ($n = 50$ for each experimental condition). **b** Upper: Schematic of experiment to measure the dependence of the observed mitotic exit delay on Mad2 and BubR1. Lower: mitotic timing in Mad2 or BubR1 RNAi treated TRIP13[AID]/APC15[KO] cells in the presence or absence of IAA. ($n = 50$ for each experimental condition). **c** Upper: Schematic of experiment to measure the dependence of the observed mitotic exit delay on Tpr. Lower: mitotic timing in Tpr RNAi treated TRIP13[AID]/APC15[KO] cells in the presence or absence of reversine and IAA. ($n = 50$ for each experimental condition). **d** Upper: Schematic of experiment to measure APC/C[MCC] formation in TRIP13[AID]/APC15[KO]. APC3 was immunoprecipitated from cells in G1 with Palbociclib and in late G2 with RO-3306. Lower: APC/C interacting BubR1 and Cdc20 levels were measured by quantitative immunoblotting. Lines in **a**–**c** represent the mean ± s.e.m.

checkpoint kinases[28–33]. A key question has been how pre-existing free and APC/C[Cdc20]-bound MCC are inactivated by disassembly or degradation so as to reactivate APC/C[Cdc20]. We show here that two separate catalytic pathways (TRIP13/p31[comet]-mediated MCC disassembly through C-Mad2 to O-Mad2 conformational conversion[47–50,52] and APC15-dependent ubiquitination of Cdc20 in MCC) are essential for inactivating MCC made either in interphase or in mitosis. Indeed, we have found that once assembled, MCC complexes are extraordinarily stable, with a negligible rate of spontaneous inactivation, producing chronic mitotic arrest in cells lacking both MCC disassembly and degradation pathways even in the absence of actively signaling kinetochores (Fig. 5). Recently, Brulotte et al.[66] showed that in vitro low levels of TRIP13 were unable to quickly reactivate APC/C[Cdc20] that had been pre-incubated with MCC components, but were sufficient to disassemble-free MCC and prevent APC/C[Cdc20] inhibition. These data clearly show that free MCC is the preferred substrate of

TRIP13, but does not eliminate the possibility of low-level activity on APC/C-MCC complexes. Thus, while our data show that both MCC disassembly pathways on their own are capable of mediating eventual mitotic exit, we postulate that in wild-type cells, a division of labor exists where TRIP13/p31[comet] preferentially disassembles free MCC, and APC15 catalyzes disassembly/degradation of APC/C-bound MCC.

Cellular studies and recent high-resolution structures of MCC-bound APC/C[Cdc20] have revealed that multiple degron sequences in the BubR1 subunit of MCC occupy the substrate-binding sites of Cdc20 to inhibit APC/C activity[41,42]. It is striking, then, that the two MCC degradation/disassembly pathways do not target BubR1 directly, instead targeting either Mad2 (in the case of TRIP13/p31[comet]) or Cdc20 (in the case of APC15-mediated ubiquitination). We have shown that while cells normally employ both pathways in parallel, either pathway is sufficient for eventual mitotic exit. These data (1) reveal that both pathways can act on APC/C[Cdc20]-bound MCC (TRIP13 can also likely act on soluble

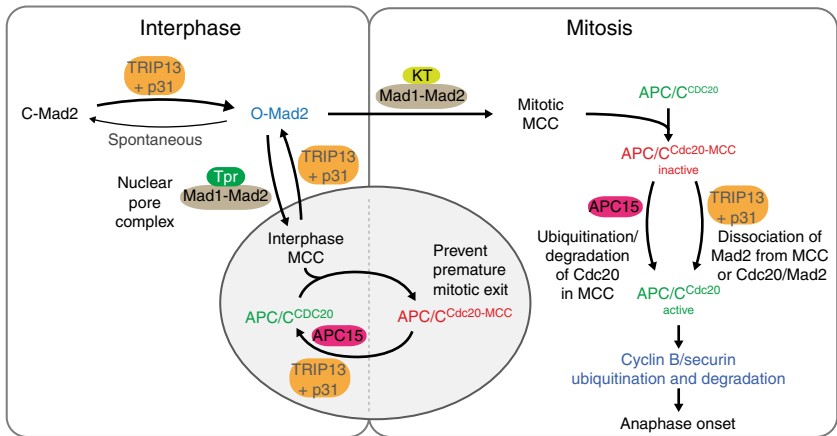

**Fig. 5** Model for TRIP13 and APC15 contributions to mitotic checkpoint signaling. In interphase cells, TRIP13 and p31comet counteract the spontaneous conversion of O-Mad2 to C-Mad2 in order to maintain a pool of O-Mad2 for checkpoint activation and regulate the homeostasis of both Mad2 and p31comet. Additionally, TRIP13 and APC15 together prevent formation of excess MCC during interphase for proper mitotic exit in the next mitosis. In prometaphase, conversion of O-Mad2 to Cdc20-bound C-Mad2 at unattached kinetochores catalyzes assembly of MCC, which binds and inhibits APC/C$^{Cdc20}$. Checkpoint silencing at metaphase can occur through two pathways: First, TRIP13 and p31comet can catalyze the disassembly of free MCC and the removal/disassembly of MCC bound to APC/C$^{Cdc20}$. Second, APC15-mediated conformational changes within the APC/C can allow ubiquitination of Cdc20 in MCC, followed by reactivation of APC/C$^{Cdc20}$

MCC) and (2) show that disruption of MCC structure by removal of either Mad2 or Cdc20 is sufficient to relieve inhibition of APC/C$^{Cdc20}$ from ubiquitination of cyclin B and securin. Earlier evidence[67] that nocodazole-treated cells depleted of TRIP13 activity (by reduction of its p31comet adapter) remain mitotically arrested after induced degradation of Mad2, but not BubR1, suggests that TRIP13's action on APC/C$^{Cdc20-MCC}$ may remove BubR1 along with Mad2 or destabilize the remaining MCC complex[22]. Indeed, BubR1 release may lag behind removal of Mad2—potentially explaining earlier observations of APC/C$^{Cdc20}$-bound "BBC" complex (BubR1, Bub3, Cdc20)[21,43,68], but it nonetheless eventually dissociates to enable APC/C$^{Cdc20}$ reactivation.

## Methods
**DNA constructs and site-directed mutagenesis**. Full-length ORFs of TRIP13 were obtained from OriGene technologies (Rockville, MD). cDNAs of TRIP13 were cloned into a pcDNA5/FRT/TO-based vector (Invitrogen) modified to contain an amino-terminus LAP tag. pCDNA5/FRT/TO-Mad2$^{WT}$ was generated as described in previous study[52]. TRIP13$^{EQ}$ variant (E253Q) was generated by QuickChangeII site-directed mutagenesis kit (Agilent).

**Cell culture and generation of stable cell lines**. All the cell lines used in this study were cultured in Dulbecco's Modified Eagle's Medium (DMEM, Thermo Fisher) supplemented with 10% tetracycline-free fetal bovine serum (Omega Scientific) and 100 U/mL penicillin-streptomycin and maintained at 37 °C under 5% CO$_2$. TRIP13$^{AID}$ cells; TIR1 and H2B$^{mRFP}$-expressing T-REx Flp-In DLD-1 cells[60] were engineered by CRISPR/Cas9 and HR-mediated genome editing to integrate green fluorescent protein (GFP) and auxin-inducible degron (AID) in front of TRIP13 locus (AID ORF was a kind gift from Masato Kanemaki; National Institute of Genetics, Mishima, Japan).

Oligonucleotides containing sgRNA sequences targeting TRIP13 are inserted into *Bbs*I site in pX330-U6-Chimeric_BB-CBh-hSpCas9 (Addgene 42230). pCR4-Blunt TOPO (Invitrogen) containing EGFP-AID cDNA with two TRIP13 homology arms (~750 bp, respectively) at both ends was constructed using Gibson assembly method (NEB). Two plasmids were co-transfected into DLD-1 TRex Flp-In TIR1-9Myc via electroporation. Two days after transfection, GFP-positive single cells were isolated using single-cell sorting (FACS-Vantage; Becton Dickinson). Targeted cells were identified through genomic PCR and DNA sequencing, further confirmed by immunoblotting. To generate TRIP13$^{AID}$/APC15$^{KO}$ cells, C11orf51 CRISPR/Cas9 KO Plasmid (Santa Cruz Biotechnology) was transfected into TRIP13$^{AID}$ cells and targeted cells were identified by immunoblotting.

For generation stable cell line inducibly expressing LAP-TRIP13$^{WT/EQ}$ and Mad2$^{WT}$-Flp, pcDNA5/FRT/TO-LAP-TRIP13 or Mad2 were co-transfected with pOG44 into TRIP13$^{AID}$ cells using X-tremeGENE 9 (Roche). Cells were selected with 150 μg/mL hygromycin B (Thermo Fisher), and protein expression was confirmed by immunoblotting.

Doxycycline and the indole-3-acetic acid (IAA) were dissolved in water and used at 1 μg/mL and 500 μM, respectively. For mitotic arrest, 0.33 μM nocodazole (Sigma Aldrich) was used. For checkpoint inactivation, 2 μM reversine (Santa Cruz Biotechnology) was used.

**RNA interference**. For depletion of endogenous proteins, TRIP13$^{AID}$ cells were transfected with 50 nM of siRNA and Lipofectamine RNAiMAX (Thermo Fisher) for 48 h (remaining) before analysis. siRNAs directed against the 3′ untranslated region of Mad2, BubR1 or On-TARGETplus-non targeting siRNA; were purchased from GE Dharmacon.

**Time-lapse live-cell imaging**. To determine mitotic timing, TRIP13$^{AID}$ cells were seeded onto CELLSTAR μClear 96-well plate (Greiner bio-one) and images were collected using a CQ1 confocal image cytometer (Yokogawa Electric Corporation) with a ×40 objective. 5 × 2-μm z-sections in RFP (20% power, 150 ms, 30% gain) were acquired in each field at 2.5, 5, or 10-min intervals. Image acquisition and data analysis were performed using ImageJ. To measure mitotic error rate, chromosome lagging and chromosome misalignment were counted during anaphase onset.

**Cell growth, flow cytometry, and clonogenic assays**. For cell doubling time measurements, cells were plated into six-well dishes in triplicate and counted at three days intervals. For flow cytometry analysis, ethanol-fixed cells were stained with 1 μg/mL Hoechst 33258 (Thermo Fisher) and Phospho Histone H3 (Ser10) antibody conjugated with Alexa647 (Cell Signaling) and analyzed with a SH800 cell sorter (Sony). For confirmation of cell-cycle synchronization, ethanol-fixed cells were stained with 10 μg/mL propidium iodide and 50 μg/mL RNase A and analyzed for DNA content by flow cytometry on a BD LSR II instrument (BD Biosciences). For clonogenic growth assays, 100 cells were plated into six-well dishes in triplicate for 21 days. Methanol-fixed colonies were stained with a 0.5% crystal violet, 25% methanol solution, and quantified using ImageJ software.

**DNA fluorescent in situ hybridization (FISH)**. For FISH, TRIP13$^{AID}$ cells were cultured in chambered slides (IBIDI) with or without IAA for 5 days and fixed in cold methanol:acetic acid (3:1) for 15 mins and dehydrated with 80% ethanol.

Centromere 4 enumeration painting probes (MetaSystems) was applied to slides, sealed with a coverslip, co-denatured at 75 °C for 2 mins, and hybridized overnight at 37 °C in a humidified chamber. Slides were subsequently washed with 0.4× SCC at 72 °C for 2 mins and rinsed in 2× SCC, 0.05% Tween-20 at room temperature for 30 s. Slides were then rinsed in water, counterstained with DAPI, and mounted in anti-fade solution. FISH images were acquired on a DeltaVision Core system (GE Health Sciences) at ×60 magnification (5 × 1μm z-sections) and maximum intensity projections were generated using softWoRx software.

**Ion exchange and size-exclusion chromatography**. For Mad2 conformation and complexes formation analyses, we prepared fresh whole-cell lysate without freeze and thaw to prevent conformation changes during preparation. For examination of Mad2 conformation, ion-exchange chromatography was carried out as described in Ye et al.[52] Whole-cell lysates were injected onto a Mono-Q column (GE

Healthcare) in buffer containing 25 mM Tris pH 7.4, 1 mM MgCl$_2$, 1 mM EDTA, 1 mM DTT and 50 mM NaCl, and then eluted with a gradient to 400 mM NaCl. For examination of Mad2 complex formation, whole-cell lysates were injected onto a Superdex 200 column (GE Healthcare). Fractions were collected and analyzed by SDS-PAGE followed by immunoblot against Mad2, p31$^{comet}$, and Mad1 antibodies.

**Antibodies and immunoprecipitation**. BubR1$^{1-322}$ was expressed in protein was expressed in *E. coli* Rosetta2 (DE3) pLysS by induction with IPTG for 20 h at 20 °C, then purified by Ni$^{2+}$ affinity (Ni-NTA; Qiagen). The protein was used to immunize a rabbit for antisera. Anti-BubR1 antibody was purified by antigen crosslinked HiTrap NHS-activated HP (GE Healthcare) column. Rabbit anti-BubR1 (homemade, 1:10000), rabbit anti-TRIP13 (Bethyl, Cat# A303–605, 1:3000), rabbit anti-Mad2 (Bethyl, Cat# A300-301A, 1:3000), mouse anti-p31$^{comet}$ (Mad2L1BP) (Santa Cruz, Cat# SC-134381, 1:200), mouse anti-Cdc20 (Santa Cruz, Cat# SC-136024, 1:200), mouse anti-APC3 (Cdc27) (Santa Cruz, Cat# SC-9972, 1:200), mouse anti-GAPDH (6C5) (Abcam, AB8245, 1:10000), and mouse anti-α-tubulin (DM1α) (Abcam, AB7291, 1:10000) were used for immunoblotting. To monitor MCC and APC/C$^{MCC}$ formation, rabbit anti-BUBR1 or mouse anti-APC3 were conjugated to M-270 epoxy Dynal bead using Dynabead antibody coupling kit (Thermo Fisher). For immunoprecipitation, conjugated antibodies were mixed with precleared whole-cell lysates that have 1 mg total protein, and rotated for 90 min at 4 °C. After three times washing with lysis buffer (20 mM Tris-HCl, pH 7.4, 150 mM NaCl, 1 mM EDTA, 0.2% NP-40), the beads were treated with 30 μL 2× SDS sample buffer and boiled for 5 min. Uncropped western blots are shown in Supplementary Figure 5.

**Protein purification**. For in vitro characterization of wild-type Mad2 or Mad2$^{H191A}$ (both of which contained the R133A mutation to reduce homo-dimerization), protein was expressed in *E. coli* Rosetta2 (DE3) pLysS by induction with IPTG for 20 h at 20 °C, then purified by Ni$^{2+}$ affinity (Ni-NTA; Qiagen) and ion-exchange (HiTrap Q HP; GE Life Sciences) chromatography. His$_6$-SUMO tags were cleaved by incubation with *S. cerevisiae* Ulp1 at 4 °C overnight, and cleaved protein was passed over a Ni$^{2+}$ affinity column a second time to remove uncleaved protein, His$_6$-SUMO tags, and Ulp1. Protein was purified by size-exclusion chromatography (Superdex 200, GE Life Sciences), concentrated by ultrafiltration, and stored at −80 °C for biochemical assays.

**Quantification and statistical analysis**. For quantification of TRIP13 and Mad2, triplicate samples were analyzed on independent Western blots, band intensity measured by ImageJ, and points plotted as mean ± standard deviation. Degradation/conformational conversion half-times were calculated in Prism v. 7 using a single-exponential decay model (plateau value set to 0% for TRIP13 degradation, but allowed to float for MAD2 conversion). For analysis of cell-cycle timing, at least 50 cells were tracked by live-cell imaging at 1–5 min intervals (depending on experiment), and are plotted as mean ± s.e.m. (standard error of the mean). For all panels showing *P*-values, these were calculated in Prism v. 7 using an unpaired two-tailed *t*-test (**$P < 0.01$, ***$P < 0.001$, NS = not significant).

## Data availability

The data that support the findings of this study are available from the authors on reasonable request, see author contributions for specific data sets.

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

## Acknowledgements

We thank Andrew Shiau and David Jenkins for sharing resources and their assistance with imaging, Arshad Desai, Pablo Lara-Gonzalez, Nam Hee Kim, members of the Cleveland and Corbett labs for helpful discussions, and acknowledge the Ludwig Institute for Cancer Research (to K.D.C. and D.W.C.), NIH R01 GM104141 (to K.D.C.), NIH K99 CA218871 (to P.L.), IBS-R022-D1-2017-a00 (to J.S.H. and K.M.), NIH R01 GM29513 (to D.W.C.), and R35 GM122476 (to D.W.C.) for funding.

## Author contributions

D.H.K., K.D.C. and D.W.C. designed the experiments. J.S.H., K.M. and M.M. generated TRIP13AID cells, D.H.K. generated TRIP13AID/APC15KO cells and performed all live-cell experiments, FACS analyses, quantitative immunoblotting, and data analysis. J.S.H. generated BubR1 antibody, P.L. performed FISH with data interpretation and Q.Y. performed Mono-Q ion-exchange chromatography. D.H.K., K.D.C. and D.W.C. wrote the manuscript.

## Additional information

**Competing interests:** The authors declare no competing interests.

