## [Peer Review File · Nature Communications]

Reviewers' comments:

Reviewer #1 (Remarks to the Author):

This manuscript describes elegant and convincing work on the mechanism of disassembly of the mitotic checkpoint complex (MCC), the mitotic checkpoint effector. The MCC targets the anaphase-promoting complex/cyclosome (APC/C) to delay Cyclin B and Securin degradation until the achievement of chromosome bi-orientation. Previous work from several laboratories has shown that removal of the MCC from the APC/C, and MCC disassembly, are related but at least partly distinct processes. One process, which requires an AAA+ ATPase named Trip13 (with a co-factor named p31comet) targets the conformation of the MCC subunit Mad2. Another process, which requires Apc15, an APC subunit, and ubiquitination, targets one of the two Cdc20 protomers in MCC, the one bound to Mad2. While this is all relatively well established, what has not been explored is how these pathways cooperate. This is important, because the phenotypes observed when depleting, individually and in various ways, Trip13, p31comet, or Apc15, are all relatively mild, and range from short delays of the metaphase-to-anaphase transition to a defect in mitotic checkpoint activation. The great merit of this paper is that it provides a very robust mechanistic framework to understand these previous observations, meaningfully combining perturbations to obtain truly dramatic phenotypes. Most of the main conclusions of the paper are well supported in my opinion. In a brief summary, these include 1) that the duration of the mitotic delay in cells lacking Trip13 depends on the amount of residual active form of Mad2 at mitotic entry; and 2) that when the Trip13 and Apc15 pathways are inactivated concomitantly, and provided sufficient MCC can be generated prior to inactivation, mitotic exit is delayed permanently even in the absence of active checkpoint signaling. Another conclusion, based on work in Figure 4, is slightly more controversial in my opinion, as clarified below.

I strongly support publication of this study, even in its present form. Nonetheless, the authors may elect to consider the following points:

-Figure 2C: Loss of Trip13 leads to a very strong accumulation of p31comet, a C-Mad2 binder. I am not asking the authors to investigate the cause of this increase, but the authors may want to consider the possibility that it acts as a stabilizing factor for C-Mad2, i.e. that the amount of C-Mad2 and the rate of conversion from O-Mad2 observed in the absence of Trip13 is influenced by the increased levels of p31comet. Of course, this does not change the substance of the observations, but provides an alternative explanation to spontaneous conversion.

-Figure 4: The potential caveat with this figure is that the analysis is carried out in the long-term absence of Apc15 and I am not sure that the levels of interphase MCC bound to APC/C are a fair representation of anything that would concern unperturbed cells. As a proof of existence of an interphase pathway to MCC accumulation, this comes across as a quite weak inference to me (even if I do believe that the interphase pathway may indeed exist, see below). All that the experiment shows is that when pathways of MCC disassembly fail, some MCC (which is never compared with the mitotic levels) is to be found on the APC/C.

-Discussion, last paragraph: the timer the authors are discussing is not related with the Meraldi et al. timer. The Meraldi et al. timer was based on an artificial distinction of checkpoint proteins in proteins that have an additional timer function and proteins that work exclusively in the checkpoint. Mad2 and BubR1 were attributed this additional function as timer components, while Mad1 was considered "exclusively" checkpoint. The timer or Meraldi et al. was invariably studied under conditions of depletion of checkpoint components, be it Mad2, BubR1, or Mad1, and it was operating post-NEBD. The authors' timer, on the other hand, operates pre-NEBD and requires checkpoint components (including Mad1) that Meraldi et al. had excluded as part of the timer. Hence, the authors' timer cannot be the same timer as that purported by Meraldi et al. (In my opinion, the Meraldi et al. timer was a mere representation of the fact that the timer subunits Mad2 and BubR1 are MCC components, and can form some MCC also in the absence of Mad1 or

Bub1, whereas depletion of Mad2 or BubR1 prevents any MCC assembly) As the authors argue that their observations are consistent with pre-NEBD production of a timer, which is sensible, the timer they refer to appears to be identical to that described by Rodrigo-Bravo et al. and previously observed by the Van Deursen laboratory in BubR1 depleted cells (for a discussion, see Musacchio, *Curr. Biol.* 2015). As already shown by Rodrigo-Bravo and Van Deursen, this timer is no less than Cyclin B itself, whose levels pre- and around NEBD are strongly reduced when the SAC is inhibited. If the checkpoint is not active, or not activated, cells entering mitosis with low levels of Cyclin B will also go out faster than cells with high levels of Cyclin B. It seems highly likely that if the pool of O-Mad2 is depleted upon interference with Trip13 function, the same problem will arise, a hypothesis that the authors could easily test.

Minor points:

-Figure 2C-D: not sure that the colored + and - serve a specific purpose, they should remain black (color-coding of gels and plots is useful, however).

-Page 11: "...continual MCC degradation..." I think that the authors mean "disassembly", not degradation.

Reviewer #2 (Remarks to the Author):

The manuscript by Kim et al investigates the role of the AAA+ ATPase TRIP13 and the APC/C subunit Apc15 in controlling the assembly and disassembly of the mitotic checkpoint complex, an effector complex of the spindle assembly checkpoint (SAC). The MCC controls anaphase onset by suppressing APC/C activity, specifically inhibiting it towards securin and cyclin B ubiquitination. Once the SAC is satisfied the APC/C-MCC complex must be disassembled and APC/C reactivated. Two mechanisms have been characterized: direct MCC disassembly by TRIP13-p31comet catalyzed conversion of C-Mad2 to O-Mad2, and APC/C-catalyzed ubiquitination of the Cdc20 subunit of the MCC. The latter process is dependent on Apc15 and results in both direct dissociation of MCC from the APC/C-MCC complex and Cdc20 degradation. MCC is assembled through a process involving the conversion of O-Mad2 to C-Mad2 that then binds Cdc20 and BubR1-Bub3. The O-Mad2 to C-Mad2 conversion occurs spontaneously but very slowly, and the conversion is catalyzed *in vivo* by unattached kinetochores at prometaphase. During interphase the O-Mad2 to C-Mad2 conversion appears to be catalyzed by proteins of the nuclear pore complex. The latter may play a role in mitotic timing. MCC assembly requires the presence of O-Mad2, thus a pool of O-Mad2 must be present for MCC assembly to occur.

In the current study the authors have developed and used an auxin-inducible based system to deplete TRIP13 and combined this with an Apc15 knockout to determine whether the TRIP13 and Apc15-dependent pathways synergize, in terms of controlling exit from mitosis and they investigate the role of TRIP13 in controlling activation of the SAC. They show that loss of TRIP13 ≥ 24 hours before the next mitosis suppresses checkpoint activation due to loss of soluble O-Mad2. In contrast, cells that enter mitosis soon after depletion of TRIP13 have a delayed mitotic exit, showing that TRIP13 controls SAC inactivation. Combining TRIP13 depletion with Apc15 depletion completely prevents mitotic exit, showing that these two SAC inactivation mechanisms are synergistic. The authors show that TRIP13 depletion leads to errors in chromosome segregation and prolonged depletion results in much smaller colony sizes. The authors show the role of TRIP13 in counter-acting the spontaneous conversion of O-Mad2 to C-Mad2.

This is an interesting and important study using a novel and effective means to control the timing of TRIP13 depletion. The experiments are well performed and robust and the results and conclusion substantially add to an understanding of the mechanism of checkpoint activation,

silencing and mitotic timing. The studies described clearly show the importance of TRIP13 in all three of these processes and in addition clarify the independent functions of TRIP13 and Apc15 in checkpoint silencing and further provide evidence that TRIP13 is important in preventing aneuploidy.

I have very minor comments the authors should consider addressing.

1. Page 8 (Fig. 2). It wasn't clear if these experiments involved an unperturbed mitosis or whether nocodazole was used.
2. A prediction of this model would be that when TRIP13 is depleted a few hours prior to mitosis, thus delaying mitotic exit, the subsequent cell cycle should not arrest in mitosis (with nocodazole). Was this tested?
3. The authors find that APC/C-MCC is assembled during interphase. Some of the experimental details are not clear. Was the APC/C-MCC isolated from a whole cell lysate or was the nucleus and cytosol fractionated? Since MCC will be present in the cytosol and APC/C in the nucleus, it isn't clear where APC/C-MCC would be generated. In addition to this APC/C-MCC requires both APC/C-Cdc20 and MCC with APC/C-Cdc20 assembly mitotic phosphorylation. Again it is not clear what levels of APC/C-Cdc20 would be present during interphase.
4. Page 12. The authors state: 'cellular O-Mad2 rapidly converted to C-Mad2 in the absence of TRIP13 with a half-time of 1.4 hours'. A half-time of 1.4 hours is not rapid. It is certainly very slow compared to the rate of conversion catalyzed by the kinetochore, as shown by ref. 17 (Faeson et al.).
5. The discussion of the mitotic timer based on TRIP13 seems a bit simplistic. Other factors would also be involved including levels of Cdc20 and Mad2 transcription etc.
6. Pages 3-5, Introduction. Some of the references are out of context and/or mis-cited. For example citing ref 1 (1994) on the APC/C which was first reported in 1995 seems strange, as is not citing Sudakin (2001) for their discovery of the MCC.

Reviewer #3 (Remarks to the Author):

Comments on Kim et al TRIP13 and APC15 drive mitotic exit by turnover of interphase- and unattached kinetochore-produced MCC

The mitotic checkpoint is an important mechanism for faithful chromosome segregation. The mitotic checkpoint monitors chromosome attachment with spindle microtubules at kinetochores, and allows anaphase onset only after the chromosomes are properly bi-oriented. Its activation depends on the formation of the mitotic checkpoint complex (MCC, composed of BUBR1, BUB3, CDC20 and MAD2). Disassembly of the MCC leads to inactivation of the mitotic checkpoint and mitotic exit.

In this manuscript, the authors interrogated the contributions of two pathways of mitotic checkpoint activation and two pathways of mitotic checkpoint silencing to a functional mitotic checkpoint. The two activation pathways produce MCC either through MAD1:MAD2 complex at nuclear pores in interphase or at unattached kinetochores in prometaphase. The two silencing pathways mediate MCC disassembly through either TRIP13 AAA-ATPase or APC15-driven CDC20 ubiquitylation.

Despite that some parts of the writing are convoluted, the experiments are well designed and implemented. Some of the results were known from recent publications such as TRIP13 knockout caused mitotic checkpoint defects. However the manuscript provided a good explanation of the dual roles of TRIP13 in both mitotic checkpoint activation and silencing. Of particular importance is that the results established that the TRIP13 and CDC20 ubiquitylation are two (and probably the

only two) major pathways for mitotic checkpoint silencing. Their results also confirmed that the MCC, once formed, is stable and needs energy input (whether it's ATPase or ubiquitylation) to be disassembled. This has been pointed out by several labs before, but they validated it using the new approach of inducible TRIP13 degradation. Because of the stability of the MCC complex, even the MCC generated from interphase cells, a minor contributor of the MCC in normal situations, could cause chronic mitotic arrest when APC15 and TRIP13 are both disrupted. This suggests that, if not disassembled, accumulation of this pool of MCC alone could override all the APC/C-CDC20 in cells. The results emphasized the dynamic balance between the mitotic checkpoint activation and silencing, which should be even more clearly discussed.

Some detailed comments are listed below that the authors should address:

Fig1E, what is the total number of cells counted?

Fig1, if TRIP13 loss compromises long-term cell fitness, how could the Poon lab and the Benezra lab establish TRIP13 knockout cell lines?

Fig2C, the MAD2WT-FIp level is different when TRIP13 is induced for degradation or not. Similarly, P31 levels also changed when TRIP13 is degraded. Why?

Fig2E, the mitotic delay was said to be caused by TRIP13 loss and checkpoint silencing defect, but it could also be the increased conversion of O-to-C with overexpressed MAD2. An O and C-MAD2 distribution as in Fig2D but in mitotic cells will be very informative.

Fig3B and 3F, during long mitotic delay, did any cells die? The percentage?

P10, second line "In interphase, Mad1:Mad2 are bound to the cytoplasmic face of the nuclear envelope". Should be "nucleoplasmic face", based on the EM result of Campbell M et al, JCS, 2000 and Rodrigues-Bravo et al 2014.

Fig4D, align APC3 bands in the "input" and IP panels.

Discussion: The correlation of interphase MCC to the "timer" for unperturbed mitosis has been pointed out at least by Rodrigues-Bravo et al or Tipton et al, Cell Cycle 2011. These references should be cited.

Brulotte et al (Nature Communications 2017) suggested that TRIP13 only disassembles free MCC not bound to APC/C. The authors suggested TRIP13 disassembles both forms of MCC. The discrepancy should be discussed.

Reviewer #4 (Remarks to the Author):

To exit from mitosis, it is important to inactivate MCC, which is produced from unattached kinetochores and nuclear pores. TRIP13, an AAA+ ATPase forming a complex with p31, is proposed to extract the MAD2 protein from MCC for inactivation of the spindle checkpoint. Paradoxically, permanent loss of TRIP13 was reported to cause a defect in the spindle checkpoint activation itself.

In this paper, Kim et al. dissected the roles of TRIP13 using a degron cell line in which degron-fused TRIP13 can be rapidly degraded. The cells depleted of TRIP13 for more than 24 h showed accelerated mitotic exit. Long-term TRIP13 depletion caused accumulation of inactive C-MAD2 in the cells, suggesting TRIP13 is important for maintaining the pool of O-MAD2 for MCC formation. These observations can be explained by the interpretation that C-MAD2 accumulated in TRIP13

depleted cells cannot be used for MCC formation causing a defect in the spindle checkpoint activation.

Detailed time-course and synchronization analyses coupled with timely removal of TRIP13 showed that TRIP13 also plays a key role in the spindle checkpoint inactivation. To compare the contribution of CDC20 degradation via APC15 for mitotic exit, the authors knocked out APC15 in the TRIP13-degrom background. The cells depleted of both APC15 and TRIP13 were completely blocked in metaphase, clearly showing that the APC15 and TRIP13 pathways function in parallel for the checkpoint inactivation leading to mitotic exit. Finally, The authors tested the contribution of MCC generated from nuclear pores in interphase and showed that it is sufficient to activate the spindle checkpoint, which is subsequently inactivated via the TRIP13 and APC15 pathways.

Technically, this is a clean paper. All cell culture experiments were carefully done by taking advantage of timely removal of TRIP13 in the degrom cell lines. The presented results are convincing, and the interphase and mitotic roles of TRIP13 were nicely dissected. In terms of conceptual advancement, many results were, to some extent, predicted or expected from previous literatures. However, it is impressive to see complete arrest of the APC15 and TRIP13 depleted cells in metaphase. This clearly indicates that inactivation of the spindle checkpoint is essential for mitotic exit.

I do not have a major concern. To improve the manuscript, I would like to point out a few minor issues.

"In both G1 and late G2, low levels of BubR1-bound Mad2 were detected in extracts from cells containing TRIP13; much higher levels were found in extracts of cells depleted of TRIP13 (by IAA induced degradation) (Figure S3D)." (page 11)

In Figure S3D, there is only a WB data of G2 cells. I could not find blots of G1 cells.

In Figure 3D, APC3-IP from G2 arrested APC15 and TRIP13 depleted extract clearly purified BUBR1, CDC20, and MAD2. However, the same IP from G1 extracts purified a significantly less amount of MAD2. Is there any explanation?

We would first like to thank the referees for their efforts and insightful comments on our initial manuscript. We feel that the revised submission is much improved, and we hope to have addressed each of the issues raised. Our comments have been added below each point in blue font.

RESPONSE TO REFEREE #1

This manuscript describes elegant and convincing work on the mechanism of disassembly of the mitotic checkpoint complex (MCC), the mitotic checkpoint effector. The MCC targets the anaphase-promoting complex/cyclosome (APC/C) to delay Cyclin B and Securin degradation until the achievement of chromosome bi-orientation. Previous work from several laboratories has shown that removal of the MCC from the APC/C, and MCC disassembly, are related but at least partly distinct processes. One process, which requires an AAA+ ATPase named Trip13 (with a co-factor named p31^{comet}) targets the conformation of the MCC subunit Mad2. Another process, which requires Apc15, an APC subunit, and ubiquitination, targets one of the two Cdc20 protomers in MCC, the one bound to Mad2. While this is all relatively well established, what has not been explored is how these pathways cooperate. This is important, because the phenotypes observed when depleting, individually and in various ways, Trip13, p31^{comet}, or Apc15, are all relatively mild, and range from short delays of the metaphase-to-anaphase transition to a defect in mitotic checkpoint activation. The great merit of this paper is that it provides a very robust mechanistic framework to understand these previous observations, meaningfully combining perturbations to obtain truly dramatic phenotypes. Most of the main conclusions of the paper are well supported in my opinion. In a brief summary, these include 1) that the duration of the mitotic delay in cells lacking Trip13 depends on the amount of residual active form of Mad2 at mitotic entry; and 2) that when the Trip13 and Apc15 pathways are inactivated concomitantly, and provided sufficient MCC can be generated prior to inactivation, mitotic exit is delayed permanently even in the absence of active checkpoint signaling. Another conclusion, based on work in Figure 4, is slightly more controversial in my opinion, as clarified below.

I strongly support publication of this study, even in its present form.

We very much appreciate the kind words from the referee.

Nonetheless, the authors may elect to consider the following points:

1. Figure 2C: Loss of Trip13 leads to a very strong accumulation of p31^{comet}, a C-Mad2 binder. I am not asking the authors to investigate the cause of this increase, but the authors may want to consider the possibility that it acts as a stabilizing factor for C-Mad2, i.e. that the amount of C-Mad2 and the rate of conversion from O-Mad2 observed in the absence of Trip13 is influenced by the increased levels of p31^{comet}. Of course, this does not change the substance of the observations, but provides an alternative explanation to spontaneous conversion.

The reviewer's observation is entirely correct that p31^{comet} levels increase dramatically upon TRIP13 degradation. Similar increases in p31^{comet} levels have been observed previously in

TRIP13-knockout cells (Ma and Poon, *Cell Reports* 2016). We also observed a drop in overall Mad2 levels when TRIP13 is degraded.

We have investigated the mechanistic basis for these changes and our results are presented in the new Figure S4: By quantitative immunoblotting, we observed a gradual reduction in Mad2 level accompanied by an increase in p31^{comet} level within several hours after TRIP13 depletion (Figure S4A, S4B). Measurement of mRNA levels by qPCR suggested that these changes occurred not through transcriptional alteration of the cells, but rather a change in TRIP13-regulated homeostasis of Mad2 and p31^{comet}. We analyzed cell lysates of TRIP13-depleted TRIP13^{AID} cells by size-exclusion chromatography and found that, as in wild-type cells, Mad2 is found predominantly as a monomer, while most p31^{comet} is in larger complexes consistent with a C-MAD2:p31^{comet} heterodimer (Figure S4C) (Mad2 levels are significantly higher than those of p31^{comet}, even considering the drop in Mad2 levels in TRIP13-depleted cells). Consistent with an overall increase in p31^{comet} levels in TRIP13-depleted cells, we observed a concomitant increase in levels of Mad2:p31^{comet} complexes when TRIP13 is depleted. This observation offers experimental support that complex formation with Mad2 stabilizes the p31^{comet} protein.

To clarify if the decrease in Mad2 levels when TRIP13 is degraded is linked to the shift from predominantly O-Mad2 to C-Mad2 after TRIP13 depletion, we engineered a variant Mad2, Mad2^{H191A} that preferentially adopts the O-Mad2 conformation in solution (Figure S4D, S4E) (first described in our previous publication: Ye et. al. *EMBO J.* 2017). We monitored the impact of TRIP13 depletion on Mad2 and p31^{comet} protein levels in cells whose endogenous Mad2 was replaced with tetracycline-inducible Mad2^{WT-Flp} or Mad2^{H191A-Flp} (Figure S4F). When TRIP13 is depleted, cells expressing Mad2^{WT-Flp} and in which the Mad2 population is a mixture of open and closed forms (Figure 2D, in brown box), a reduction in Mad2 levels and an increase in p31^{comet} level was seen, just as was as seen in cells with endogenous Mad2 (Figure S4F). Interestingly, cells expressing Mad2^{H191A-Flp} showed no measurable change in either Mad2 or p31^{comet} levels after TRIP13 depletion, even 24 hours after new Mad2 synthesis was blocked by washing out doxycycline (Figure S4F). This result suggests that homeostasis of Mad2 and p31^{comet} depends strongly on Mad2 conformation. Based on the reduction in Mad2 level upon TRIP13 depletion in cells expressing Mad2^{WT}, we also hypothesize that the turnover rate of C-Mad2 in cells is significantly faster than the turnover of O-Mad2.

We agree with the reviewer's point that p31^{comet} likely stabilizes Mad2 in its closed conformation in solution. The results above, however, point to a more complex homeostatic mechanism that will require careful dissection in future studies.

2. Figure 4: The potential caveat with this figure is that the analysis is carried out in the long-term absence of Apc15 and I am not sure that the levels of interphase MCC bound to APC/C are a fair representation of anything that would concern unperturbed cells. As a proof of existence of an interphase pathway to MCC accumulation, this comes across as a quite weak inference to me (even if I do believe that the interphase pathway may indeed exist, see below). All that the experiment shows is that when pathways of MCC disassembly fail, some MCC (which is never compared with the mitotic levels) is to be found on the APC/C.

The referee is completely correct on these points. Our data indicate that cells accumulate MCC during interphase when both MCC disassembly/degradation pathways are compromised, but does not shed light on whether MCC can accumulate in interphase when one or both of these pathways are active. While the manuscript text in the results section accurately notes these caveats, we have now altered the discussion (see next point) to more accurately describe what we can and cannot infer from these data.

3. Discussion, last paragraph: the timer the authors are discussing is not related with the Meraldi et al. timer. The Meraldi et al. timer was based on an artificial distinction of checkpoint proteins in proteins that have an additional timer function and proteins that work exclusively in the checkpoint. Mad2 and BubR1 were attributed this additional function as timer components, while Mad1 was considered “exclusively” checkpoint. The timer of Meraldi et al. was invariably studied under conditions of depletion of checkpoint components, be it Mad2, BubR1, or Mad1, and it was operating post-NEBD. The authors’ timer, on the other hand, operates pre-NEBD and requires checkpoint components (including Mad1) that Meraldi et al. had excluded as part of the timer. Hence, the authors’ timer cannot be the same timer as that purported by Meraldi et al. (In my opinion, the Meraldi et al. timer was a mere representation of the fact that the timer subunits Mad2 and BubR1 are MCC components, and can form some MCC also in the absence of Mad1 or Bub1, whereas depletion of Mad2 or BubR1 prevents any MCC assembly) As the authors argue that their observations are consistent with pre-NEBD production of a timer, which is sensible, the timer they refer to appears to be identical to that described by Rodrigo-Bravo et al. and previously observed by the Van Deursen laboratory in BubR1 depleted cells (for a discussion, see Musacchio, *Curr. Biol.* 2015). As already shown by Rodrigo-Bravo and Van Deursen, this timer is no less than Cyclin B itself, whose levels pre- and around NEBD are strongly reduced when the SAC is inhibited. If the checkpoint is not active, or not activated, cells entering mitosis with low levels of Cyclin B will also go out faster than cells with high levels of Cyclin B. It seems highly likely that if the pool of O-Mad2 is depleted upon interference with Trip13 function, the same problem will arise, a hypothesis that the authors could easily test.

Upon further consideration of both our data and the data cited by the reviewer above, we agree with the reviewer. We conclude that while our data provide support for interphase generation of MCC, they do not significantly contribute to the models of the mitotic timer proposed either by Meraldi or Rodriguez-Bravo/Van Deursen. Therefore, we have removed the paragraph in question, and confined our discussion to the confirmation of interphase MCC production (Discussion, paragraph 2).

Minor points:

1. Figure 2C-D: not sure that the colored + and - serve a specific purpose, they should remain black (color-coding of gels and plots is useful, however).

While our intention with the color-coded + and – symbols was to provide a visual link between different panels of the figure, we now recognize that this may have been more confusing than clarifying for a reader. We have taken the reviewer’s suggestion that these symbols remain black. We have also added color-coded lines above each lane of the blot in Figure 2C to provide

a visual link with the same conditions in Figure 2D and 2E, which are each equivalently color-coded.

2. Page 11: "...continual MCC degradation..." I think that the authors mean "disassembly", not degradation.

We thank the referee for catching this error, and we have corrected the text accordingly.

RESPONSE TO REFEREE #2

The manuscript by Kim et al investigates the role of the AAA+ ATPase TRIP13 and the APC/C subunit Apc15 in controlling the assembly and disassembly of the mitotic checkpoint complex, an effector complex of the spindle assembly checkpoint (SAC). The MCC controls anaphase onset by suppressing APC/C activity, specifically inhibiting it towards securin and cyclin B ubiquitination. Once the SAC is satisfied the APC/C-MCC complex must be disassembled and APC/C reactivated. Two mechanisms have been characterized: direct MCC disassembly by TRIP13-p31comet catalyzed conversion of C-Mad2 to O-Mad2, and APC/C-catalyzed ubiquitination of the Cdc20 subunit of the MCC. The latter process is dependent on Apc15 and results in both direct dissociation of MCC from the APC/C-MCC complex and Cdc20 degradation. MCC is assembled through a process involving the conversion of O-Mad2 to C-Mad2 that then binds Cdc20 and BubR1-Bub3. The O-Mad2 to C-Mad2 conversion occurs spontaneously but very slowly, and the conversion is catalyzed in vivo by unattached kinetochores at prometaphase. During interphase the O-Mad2 to C-Mad2 conversion appears to be catalyzed by proteins of the nuclear pore complex. The latter may play a role in mitotic timing. MCC assembly requires the presence of O-Mad2, thus a pool of O-Mad2 must be present for MCC assembly to occur.

In the current study the authors have developed and used an auxin-inducible based system to deplete TRIP13 and combined this with an Apc15 knockout to determine whether the TRIP13 and Apc15-dependent pathways synergize, in terms of controlling exit from mitosis and they investigate the role of TRIP13 in controlling activation of the SAC. They show that loss of TRIP13 ≥ 24 hours before the next mitosis suppresses checkpoint activation due to loss of soluble O-Mad2. In contrast, cells that enter mitosis soon after depletion of TRIP13 have a delayed mitotic exit, showing that TRIP13 controls SAC inactivation. Combining TRIP13 depletion with Apc15 depletion completely prevents mitotic exit, showing that these two SAC inactivation mechanisms are synergistic. The authors show that TRIP13 depletion leads to errors in chromosome segregation and prolonged depletion results in much smaller colony sizes. The authors show the role of TRIP13 in counter-acting the spontaneous conversion of O-Mad2 to C-Mad2.

This is an interesting and important study using a novel and effective means to control the timing of TRIP13 depletion. The experiments are well performed and robust and the results and conclusion substantially add to an understanding of the mechanism of checkpoint activation, silencing and mitotic timing. The studies described clearly show the importance of TRIP13 in all three of these processes and in addition clarify the independent functions of TRIP13 and Apc15

in checkpoint silencing and further provide evidence that TRIP13 is important in preventing aneuploidy.

We very much appreciate the kind words from the referee.

I have very minor comments the authors should consider addressing.

1. Page 8 (Fig. 2). It wasn't clear if these experiments involved an unperturbed mitosis or whether nocodazole was used.

We apologize for not explaining these experiments more clearly. All the results in Figure 2 except panel B represent unperturbed mitotic timing of randomly cycling cells. Cells in Figure 2B are synchronized at G1 by Palbociclib, as indicated in the schematic above the panel. The experiments done with nocodazole treatment were Figure 1C, 3B and S1E; the schematics for these experiments all show the addition of Nocodazole as "Noc".

2. A prediction of this model would be that when TRIP13 is depleted a few hours prior to mitosis, thus delaying mitotic exit, the subsequent cell cycle should not arrest in mitosis (with nocodazole). Was this tested?

Yes, and we found just what the reviewer has correctly predicted - that any cell entering mitosis more than ~6 hours after depletion of TRIP13 does not arrest in mitosis with nocodazole treatment. Ma and Poon (*Cell Reports* 2016) previously showed that TRIP13-knockout cells are unable to arrest in mitosis with nocodazole treatment. We followed cells for two cell cycles after TRIP13 depletion without nocodazole treatment, and observed broadly similar results. We found that cells were strongly delayed in exiting mitosis within the first 3 hours after TRIP13 degradation, but these cells' daughters showed faster-than-wildtype exit from mitosis. This indicates that the daughter cells likely did not activate the checkpoint for even the short time that most wild-type cells do (see Figure at right – dotted line at ~20 minutes shows wild-type metaphase-to-anaphase timing).

We have added these new data to Fig. S2D.

3. The authors find that APC/C-MCC is assembled during interphase. Some of the experimental details are not clear. Was the APC/C-MCC isolated from a whole cell lysate or was the nucleus and cytosol fractionated? Since MCC will be present in the cytosol and APC/C in the nucleus, it isn't clear where APC/C-MCC would be generated. In addition to this APC/C-MCC requires both APC/C-Cdc20 and MCC with APC/C-Cdc20 assembly mitotic phosphorylation. Again it is not clear what levels of APC/C-Cdc20 would be present during interphase.

We apologize for the lack of clarity on these points. Our experiments were done with whole-cell lysate. Current models posit that interphase MCC is produced by Mad1:Mad2 complexes localized on the nucleoplasmic face of nuclear pore complexes, where this complex localizes in interphase (Campbell M et al, *JCS*, 2000 and Rodrigues-Bravo et al, *Cell*, 2014.). In this model, both APC/C and MCC would be nuclear. The reviewer's second point about the levels of APC/C-Cdc20 in interphase is also well-taken. While we cannot comment on the fraction of

APC/C that is bound to Cdc20 in interphase, we do detect Cdc20 in APC3 immunoprecipitations in both G1 and G2 (Figure 4D).

4. Page 12. The authors state: 'cellular O-Mad2 rapidly converted to C-Mad2 in the absence of TRIP13 with a half-time of 1.4 hours'. A half-time of 1.4 hours is not rapid. It is certainly very slow compared to the rate of conversion catalyzed by the kinetochore, as shown by ref. 17 (Faeson et al.).

We have removed the word "rapidly" from this sentence.

5. The discussion of the mitotic timer based on TRIP13 seems a bit simplistic. Other factors would also be involved including levels of Cdc20 and Mad2 transcription etc.

We agree with the reviewer on this point. In response to both this question and the extended point of reviewer #1 above, we have removed most of our discussion of the mitotic timer from the revised manuscript. The mitotic timer, in the sense of a pool of MCC that pre-exists upon mitotic entry to inhibit premature mitotic exit prior to full SAC activation, will depend on the interplay between many factors, including the levels of each component as the reviewer notes, but also on the relative rates of MCC assembly and disassembly at each cell-cycle stage.

6. Pages 3-5, Introduction. Some of the references are out of context and/or mis-cited. For example citing ref 1 (1994) on the APC/C which was first reported in 1995 seems strange, as is not citing Sudakin (2001) for their discovery of the MCC.

We thank the referee for catching the errors and we have corrected the references accordingly.

RESPONSE TO REFEREE #3

The mitotic checkpoint is an important mechanism for faithful chromosome segregation. The mitotic checkpoint monitors chromosome attachment with spindle microtubules at kinetochores, and allows anaphase onset only after the chromosomes are properly bi-oriented. Its activation depends on the formation of the mitotic checkpoint complex (MCC, composed of BUBR1, BUB3, CDC20 and MAD2). Disassembly of the MCC leads to inactivation of the mitotic checkpoint and mitotic exit.

In this manuscript, the authors interrogated the contributions of two pathways of mitotic checkpoint activation and two pathways of mitotic checkpoint silencing to a functional mitotic checkpoint. The two activation pathways produce MCC either through MAD1:MAD2 complex at nuclear pores in interphase or at unattached kinetochores in prometaphase. The two silencing pathways mediate MCC disassembly through either TRIP13 AAA-ATPase or APC15-driven CDC20 ubiquitylation.

Despite that some parts of the writing are convoluted, the experiments are well designed and implemented. Some of the results were known from recent publications such as TRIP13 knockout caused mitotic checkpoint defects. However the manuscript provided a good explanation of the dual roles of TRIP13 in both mitotic checkpoint activation and silencing. Of particular importance is that the results established that the TRIP13 and CDC20 ubiquitylation

are two (and probably the only two) major pathways for mitotic checkpoint silencing. Their results also confirmed that the MCC, once formed, is stable and needs energy input (whether it's ATPase or ubiquitylation) to be disassembled. This has been pointed out by several labs before, but they validated it using the new approach of inducible TRIP13 degradation. Because of the stability of the MCC complex, even the MCC generated from interphase cells, a minor contributor of the MCC in normal situations, could cause chronic mitotic arrest when APC15 and TRIP13 are both disrupted. This suggests that, if not disassembled, accumulation of this pool of MCC alone could override all the APC/C-CDC20 in cells. The results emphasized the dynamic balance between the mitotic checkpoint activation and silencing, which should be even more clearly discussed.

Some detailed comments are listed below that the authors should address:

1. Fig1E, what is the total number of cells counted?

We apologize for omitting this detail. We performed 3 independent experiments for Fig1E. The number of counted cells for untreated control experiments were 149, 561 and 446. The number of counted cells of IAA treatment experiments were 162, 479 and 478. We have added these cell counts to the legend for Figure 1E.

2. Fig1, if TRIP13 loss compromises long-term cell fitness, how could the Poon lab and the Benezra lab establish TRIP13 knockout cell lines?

Prior studies have shown that cells from patients with biallelic *Bub1B* or *TRIP13* mutations cause checkpoint defects resulting in mosaic variegated aneuploidy (MVA) syndrome exhibit checkpoint defects (Hanks, S., Nat. Genet., 2004, Yost et al., Nat. Genet., 2017), suggesting that a functioning mitotic checkpoint is not essential for survival. Notably, most of MVA patients showed clinical characteristics including microcephaly and growth retardation (Callier et al., Am J Med Genet A, 2005). It has also been reported that a mouse *TRIP13* gene-trap allele that results in a severe drop in protein levels causes partially-penetrant lethality, revealed by homozygote mice were being born at about 2/3 of the expected ratio (Li and Schmenti, PLoS Genet., 2007). Our results are broadly consistent with these findings: while long-term cell fitness is clearly lower in the absence of TRIP13 (Figure S1H), our colony formation assays show that growth is only reduced by ~2-fold. Thus, TRIP13 knockout cells, while they likely accumulate aneuploidy (resulting in a fraction of cells dying in each generation), are nonetheless viable.

3. Fig2C, the MAD2WT-Flp level is different when TRIP13 is induced for degradation or not. Similarly, P31 levels also changed when TRIP13 is degraded. Why?

The reviewer hits on an important point here, namely that both p31^{comet} and Mad2 levels change in the absence of TRIP13. This is not a new observation, having been reported in both TRIP13-knockout cell lines and patient cells with biallelic TRIP13 mutations (Ma and Poon, 2016, Yost et al., 2017). Given our ability to quickly degrade TRIP13, we are well-positioned to characterize this phenomenon more fully. As such, **we have added a new figure, Figure S4, to the manuscript.** This figure is described fully in our response to Reviewer #1 (point #1) above. To summarize, we find that p31^{comet} is stabilized upon TRIP13 degradation, likely through binding to

C-Mad2. The lower Mad2 levels observed upon TRIP13 degradation, on the other hand, appear to depend directly on its spontaneous conversion to C-Mad2, which is more rapidly degraded than C-Mad2. When we replaced Mad2 with a mutant (Mad2^{H191A}) that preferentially adopts the O-Mad2 conformation, we no longer observed a drop in Mad2 levels upon TRIP13 degradation.

4. Fig2E, the mitotic delay was said to be caused by TRIP13 loss and checkpoint silencing defect, but it could also be the increased conversion of O-to-C with overexpressed MAD2. An O and C-MAD2 distribution as in Fig2D but in mitotic cells will be very informative.

The reviewer is correct that logically, an increase in Mad2 levels (with concomitant increase in spontaneous O-to-C conversion) might be expected to alter mitotic timing on its own. We have previously monitored mitotic timing and Mad2 conformation with overexpressed Mad2 in the presence of TRIP13 (Ye et al 2017), and found that mitotic timing was not significantly altered. Further, our results in Figure 4 show that the mitotic delay induced by TRIP13 degradation is dependent on both Mps1 kinase activity and BubR1, demonstrating that it is a bona fide checkpoint response that causes this delay.

5. Fig3B and 3F, during long mitotic delay, did any cells die? The percentage?

We excluded dead cell during mitosis and the results only represent cells that were alive at the end of filming. However, we observed all the cells arrested in metaphase finally died after showing cohesion fatigue. In Figure 3F, round-shaped condensed chromosomes represent chromosomes starting cohesion fatigue.

6. P10, second line “In interphase, Mad1:Mad2 are bound to the cytoplasmic face of the nuclear envelope”. Should be “nucleoplasmic face”, based on the EM result of Campbell M et al, JCS, 2000 and Rodrigues-Bravo et al 2014.

We thank the referee for catching the errors and we have corrected the text accordingly.

7. Fig4D, align APC3 bands in the “input” and IP panels.

We thank the referee for catching the errors and we have corrected the figure accordingly.

8. Discussion: The correlation of interphase MCC to the “timer” for unperturbed mitosis has been pointed out at least by Rodrigues-Bravo et al or Tipton et al, Cell Cycle 2011. These references should be cited.

As noted in above responses to both reviewers #1 and #2, we now recognize that our discussion of the mitotic timer was somewhat simplistic in the originally-submitted manuscript. In the revised manuscript, we confine the discussion to the idea that TRIP13 and APC15 counteract MCC assembly in both interphase and mitosis, eschewing explicit discussion of a mitotic timer. As suggested by the reviewer, we now cite Rodrigues-Bravo et al (Cell 2014) when discussing interphase MCC generation.

9. Brulotte et al (Nature Communications 2017) suggested that TRIP13 only disassembles free MCC not bound to APC/C. The authors suggested TRIP13 disassembles both forms of MCC. The discrepancy should be discussed.

This is an excellent point, and one that we have considered carefully. Brulotte et al. (2017) tested whether TRIP13–p31^{comet} could inactivate APC/C-MCC in vitro, using the ubiquitination activity of APC/C as a readout. Their data shows that TRIP13–p31^{comet} was mostly unable to relieve the inhibition of APC/C^{Cdc20} that was pre-bound to MCC. These data suggest that free MCC is probably much more amenable to disassembly by TRIP13 than is APC/C-bound MCC, a point with which we would agree completely.

That said, in our opinion, while Brulotte has clearly shown that free MCC is a better substrate for TRIP13, they have not proven that TRIP13 is completely incapable of disassembling APC-C bound MCC. We have previously shown that TRIP13 is a relatively slow enzyme, and the low levels of TRIP13 (50 nM, meaning only ~8 nM active TRIP13 hexamer) relative to MCC components (600 nM – 1 uM) used by Brulotte et al was likely just enough to detect the more robust activity on free MCC.

Our cell-based data measuring mitotic timing, meanwhile, cannot distinguish between disassembly of free versus APC/C-bound MCC. We can say only that cells do exit mitosis when APC15-mediated MCC ubiquitination is eliminated. Therefore, TRIP13 and p31^{comet} are able, at some level, to disassemble APC/C-bound MCC.

We recognize the importance of clearly discussing these points. Therefore, we have added a short discussion of the discrepancy between our data and that of Brulotte et al to the Discussion (page 13).

RESPONSE TO REFEREE #4

To exit from mitosis, it is important to inactivate MCC, which is produced from unattached kinetochores and nuclear pores. TRIP13, an AAA+ ATPase forming a complex with p31, is proposed to extract the MAD2 protein from MCC for inactivation of the spindle checkpoint. Paradoxically, permanent loss of TRIP13 was reported to cause a defect in the spindle checkpoint activation itself.

In this paper, Kim et al. dissected the roles of TRIP13 using a degron cell line in which degron-fused TRIP13 can be rapidly degraded. The cells depleted of TRIP13 for more than 24 h showed accelerated mitotic exit. Long-term TRIP13 depletion caused accumulation of inactive C-MAD2 in the cells, suggesting TRIP13 is important for maintaining the pool of O-MAD2 for MCC formation. These observations can be explained by the interpretation that C-MAD2 accumulated in TRIP13 depleted cells cannot be used for MCC formation causing a defect in the spindle checkpoint activation.

Detailed time-course and synchronization analyses coupled with timely removal of TRIP13 showed that TRIP13 also plays a key role in the spindle checkpoint inactivation. To compare the contribution of CDC20 degradation via APC15 for mitotic exit, the authors knocked out APC15

in the TRIP13-degron background. The cells depleted of both APC15 and TRIP13 were completely blocked in metaphase, clearly showing that the APC15 and TRIP13 pathways function in parallel for the checkpoint inactivation leading to mitotic exit. Finally, The authors tested the contribution of MCC generated from nuclear pores in interphase and showed that it is sufficient to activate the spindle checkpoint, which is subsequently inactivated via the TRIP13 and APC15 pathways.

Technically, this is a clean paper. All cell culture experiments were carefully done by taking advantage of timely removal of TRIP13 in the degron cell lines. The presented results are convincing, and the interphase and mitotic roles of TRIP13 were nicely dissected. In terms of conceptual advancement, many results were, to some extent, predicted or expected from previous literatures. However, it is impressive to see complete arrest of the APC15 and TRIP13 depleted cells in metaphase. This clearly indicates that inactivation of the spindle checkpoint is essential for mitotic exit.

We thank the referee for these points and for acknowledging the sophistication of the system we developed.

I do not have a major concern. To improve the manuscript, I would like to point out a few minor issues.

1. "In both G1 and late G2, low levels of BubR1-bound Mad2 were detected in extracts from cells containing TRIP13; much higher levels were found in extracts of cells depleted of TRIP13 (by IAA induced degradation) (Figure S3D)." (page 11) In Figure S3D, there is only a WB data of G2 cells. I could not find blots of G1 cells.

We regret the confusion on this point. We thank the referee for catching the errors and we have corrected the text accordingly:

"In both G1 and late G2, low levels of APC/C-associated BubR1 and Mad2 were detected in extracts from cells containing TRIP13; much higher levels were found in extracts of cells depleted of TRIP13 (by IAA induced degradation) (Figure 4D). Overall MCC levels (as measured by BubR1-bound Mad2) also increased upon degradation of TRIP13 in G2-arrested cells (Figure S3D)."

2. In Figure 3D, APC3-IP from G2 arrested APC15 and TRIP13 depleted extract clearly purified BUBR1, CDC20, and MAD2. However, the same IP from G1 extracts purified a significantly less amount of MAD2. Is there any explanation?

The reviewer is absolutely correct that there is a striking difference in the levels of APC/C-associated BubR1 and Mad2 in G1 compared to late G2. We can envision two potential reasons for this difference. First, TRIP13 is cell-cycle regulated, having been previously shown to co-expressed with a number of mitotic proteins (Tipton et. al. *BMC Cell Biology* 2012). TRIP13 levels may therefore be higher in G1 than in late G2, leading to more robust disassembly of APC/C-MCC. Second, as APC/C is known to bind Cdh1 in late mitosis/G1, there may simply be

more APC/C-Cdc20, which is capable of binding MCC, in late G2 compared to G1. We have not updated the text to deal with these points - as they are highly speculative at this point.

REVIEWERS' COMMENTS:

Reviewer #1 (Remarks to the Author):

I thank the authors for their effort in responding to the reviewers' comments. I strongly support publication of this important manuscript in its current form.

Reviewer #2 (Remarks to the Author):

The authors have satisfactorily addressed all comments and I recommend publication of this interesting paper in Nat Comms.

Reviewer #3 (Remarks to the Author):

The authors have addressed my concerns and I think the main results are largely reconciled with earlier observations now with mechanistic explanations.
Just one minor point: the new Fig S4 was not mentioned at all in the main text.

We would first like to thank the referees for their efforts and insightful comments on our revised manuscript. We feel that the final submission is much improved, and we hope to have addressed all points raised by the reviewers. Our comments have been added below each point in **blue** font.

RESPONSE TO REFEREE #1

I thank the authors for their effort in responding to the reviewers' comments. I strongly support publication of this important manuscript in its current form.

We very much appreciate the kind words from the referee.

RESPONSE TO REFEREE #2

The authors have satisfactorily addressed all comments and I recommend publication of this interesting paper in *Nat. Comms*.

We very much appreciate the kind words from the referee.

RESPONSE TO REFEREE #3

The authors have addressed my concerns and I think the main results are largely reconciled with earlier observations now with mechanistic explanations.

We very much appreciate the kind words from the referee.

Just one minor point: the new Fig S4 was not mentioned at all in the main text.

We apologize for not explaining these experiments more clearly. To make our manuscript clearer, we have first switched the order of Supplementary Figures 3 and 4 and added a short explanation in the main text regarding Supplementary Figure 3 (originally Supplementary Figure 4).